# Network-to-Network Regularization: Enforcing Occam's Razor to Improve Generalization

**Rohan Ghosh and Mehul Motani**
Department of Electrical and Computer Engineering
N.1 Institute for Health
Institute of Data Science
National University of Singapore
rghosh92@gmail.com, motani@nus.edu.sg

## Abstract

What makes a classifier have the ability to generalize? There have been a lot of important attempts to address this question, but a clear answer is still elusive. Proponents of complexity theory find that the complexity of the classifier's function space is key to deciding generalization, whereas other recent work reveals that classifiers which extract invariant feature representations are likely to generalize better. Recent theoretical and empirical studies, however, have shown that even within a classifier's function space, there can be significant differences in the ability to generalize. Specifically, empirical studies have shown that among functions which have a good training data fit, functions with lower Kolmogorov complexity (KC) are likely to generalize better, while the opposite is true for functions of higher KC. Motivated by these findings, we propose, in this work, a novel measure of complexity called Kolmogorov Growth (KG), which we use to derive new generalization error bounds that only depend on the final choice of the classification function. Guided by the bounds, we propose a novel way of regularizing neural networks by constraining the network trajectory to remain in the low KG zone during training. Minimizing KG while learning is akin to applying the Occam's razor to neural networks. The proposed approach, called network-to-network regularization, leads to clear improvements in the generalization ability of classifiers. We verify this for three popular image datasets (MNIST, CIFAR-10, CIFAR-100) across varying training data sizes. Empirical studies find that conventional training of neural networks, unlike network-to-network regularization, leads to networks of high KG and lower test accuracies. Furthermore, we present the benefits of N2N regularization in the scenario where the training data labels are noisy. Using N2N regularization, we achieve competitive performance on MNIST, CIFAR-10 and CIFAR-100 datasets with corrupted training labels, significantly improving network performance compared to standard cross-entropy baselines in most cases. These findings illustrate the many benefits obtained from imposing a function complexity prior like Kolmogorov Growth during the training process.

## 1 Introduction and Motivation

On the surface, the problem of learning to generalize well over unseen data seems an impossible task. Classification is inherently a problem in function estimation, and the finite information that the training data samples impart seems hardly enough to be able to correctly guess the behaviour of the function over the unseen data samples outside the training set. However, assumption of a structured ground truth label function, which is the unknown function which generates the ground truth label for any datapoint, leads to more optimistic outlook on the problem. Without the assumption of

35th Conference on Neural Information Processing Systems (NeurIPS 2021).

structuredness in the ground truth, learning is not guaranteed, as was observed in the no free lunch theorem [1]. As shown in that work, there is no universal learning algorithm which can generalize well for possible choices of ground truth label functions.

Since one does not have any control over the true nature of the ground truth function in any classification problem, the other important parameter that decides the ability of a classifier to generalize, is the complexity of its function space itself. Over the past few decades, there have been multiple attempts at bounding the generalization error on the basis of various complexity measures [2, 3, 4] of the classifier's function space $\mathcal{F}$. The overall results of these theoretical developments indicate that generalization is primarily governed by metrics that are proportional to the *size* of $\mathcal{F}$. Examples of complexity metrics in this regard include Rademacher complexity, VC-Dimension, and Local forms of Rademacher Complexity (for more tight bounds). However, as observed in [5] these metrics lead to relatively loose bounds on the generalization performance for deep neural networks. This is because deep neural networks have high complexity spaces with flexibility to learn any label assignment on any set of training datapoints. This was perhaps most clearly observed in [6], where even after random label assignments on the training data samples, the networks still were able to fit the training data labels with no errors. Thus, often other methods have been explored to establish the relationship between complexity measures and generalization performance [7, 8], such as causal relations.

These studies lead to a natural question, which is, other than the metrics which relate to the size of the function space of a classifier, what other factors contribute towards its ability to generalize? State-of-the-art deep neural networks, as observed above, will yield very high complexity spaces, which does not explain their remarkable ability to generalize well in complex high-dimensional supervised classification problems, such as in vision. We note, that metrics such as Rademacher complexity or VC dimension, which relate to whole function spaces $\mathcal{F}$ (or subspaces within $\mathcal{F}$ which fit the training data well), essentially assign the same generalization gap to all functions within $\mathcal{F}$ (or the subspace within $\mathcal{F}$). However, there has been a longstanding understanding of the fact that usually among all functions which fit the training data well, simpler functions are expected to generalize better. This is an example of the Occam's Razor principle, which states that among all hypotheses that explain a phenomenon, the simplest hypothesis is preferred [9]. In the context of neural networks, even for very high complexity deep neural network function spaces, there will always be some weight configurations of a deep neural network, which yield *simpler* functions. For example, all network weight configurations yielding input-output functions which can still be efficiently approximated by smaller, shallower networks, could be considered to be *simpler*. An extreme example of this would be where we assign all weight values to zero within a deep neural network. The function that results from this weight configuration is essentially the constant function $f(X) = 0$. This observation points to the fact that even within a function space of a deep CNN, not all functions are *equally complex*, as some of them can still be approximated by shallower networks.

To that end, a primary objective of this paper is to probe complexity measures which enable us to assign a level of complexity to individual functions within a function space. Subsequently, our objective would be to steer the learning process towards network configurations which yield less complex input-output functions. An important work which explored similar directions is [10], where metrics from algorithmic complexity were developed to bias the learning process towards simpler functions. In this regard, measures of *descriptional complexity* [5] of functions have been proposed over the years, which quantify the level of complexity of an individual function, based on its shortest description. Although fundamental measures of descriptional complexity such as Kolmogorov Complexity [11] or Solomonoff Probability [12] are uncomputable, computable approximations to them have been developed [13]. In spite of this early work, there is a lack of concrete theoretical or empirical work to continue this line of investigation after [10]. Mainly, there is a lack of work that explores the relevance of such descriptional complexity measures to the generalizational ability of functions. That is, until recently, when interest in the descriptional complexity of a function was renewed as it was found that outputs of *random* maps tend to be biased towards simpler functions [14]. This result indirectly hinted that for classification tasks, the ground truth labelling function has a simple description with high probability. This was further investigated in [5] in the context of deep neural networks, where it was found that the Lempel-Ziv (*LZ*) complexity (a form of descriptional complexity) of most neural networks with random choices of weights are low (shorter description), in accordance with the result in [14]. Empirically, it was found that network weight configurations which lead to functions of smaller Lempel-Ziv complexity show better generalization performance.

These results and observations show that descriptional complexity measures may help to understand empirically the generalization behaviour of deep neural networks. In this paper, we advance this line of investigation by proposing a new theoretical framework for relating descriptional complexity measures to generalization performance. Subsequently, we also provide a computable method which improves generalization performance of neural networks by lowering their descriptional complexity.

## 2 Contributions

This paper makes the following contributions.

1. First, we undertake a brief theoretical analysis for exploring the relevance of descriptional complexity measures of functions to their expected generalization error. We propose a novel measure of complexity called *Kolmogorov Growth* ($KG$). Error bounds are estimated which portray the dependence of generalization error of a single function $f$ to $KG(f)$. The bounds depict that functions of higher $KG(f)$ will likely lead to a higher generalization gap. This result formalizes the Occam's razor principle for classifiers, and also concurs with the empirical findings in [5], where functions of higher *LZ*-Complexity showed worse generalization performance.

2. Like Kolmogorov Complexity, $KG$ is also uncomputable. Therefore, we propose computable approximations of $KG$ for neural networks, based on the concept of teacher-student approximation (similar to knowledge distillation [15]). Specifically, we show that neural network functions $f$ which can be approximated well by smaller networks will have smaller empirical $KG$ with high probability.

3. Next, using this idea, we then develop a novel method for regularizing neural networks, called network-to-network (N2N) regularization. N2N regularization forces trained network configurations to be of low $KG$. We find that doing so not only improves the generalization performance across a range of training data sizes, but also helps in the case of label noise. For instance in MNIST, we see test error decrease by 94%, reaching results competitive with benchmark methods.

4. Finally, we study the evolution of empirical $KG$ as networks are trained and observe that, in the standard classification scenario, networks show a sharp decrease in their $KG$ as training progresses. However, this trend completely reverses when the training data labels are noise corrupted and N2N regularization is able to stem the undesirable increase of $KG$ during training.

## 3 Kolmogorov Growth: Relevance to Generalization

Assume that our training data consists of $m$ $d$-dimensional i.i.d samples and their labels $S = [(z_1, y_1), (z_2, y_2), .., (z_m, y_m)]$ drawn from some distribution $P^m$.

We propose two growth measures for a single function $f$, namely Kolmogorov Growth $KG_m(f)$ and empirical Kolmogorov Growth $\widehat{KG}_S(f)$. Note that, when appropriate, we may drop the subscript $m$ and refer to $KG_m(f)$ as $KG(f)$. These measures are primarily motivated from the well-known growth function in statistics. The growth function $\Pi_m(\mathcal{F})$ is defined as

$$\Pi_m(\mathcal{F}) = \max_{\mathbf{z} \in \mathcal{R}^d} |\{(f(z_1), f(z_2), ..., f(z_m)) : f \in \mathcal{F}\}| \tag{1}$$

Note that the growth function is defined for a space of functions and captures the maximum number of label assignments a function space $\mathcal{F}$ can generate on any $m$ points in $\mathbb{R}^d$.

To define Kolmogorov Growth measures for a single function $f$, we first need to generate a function space from $f$, based on a description $D_f$ of $f$. This is done via noting all the parameters involved in the description $D_f$ (other than the co-ordinates in $\mathbb{R}^d$), and then assigning all possible values to those parameters to generate a space of functions $\mathcal{F}(D_f)$ from the description $D_f$. Next, we note that for a function $f$, we consider multiple descriptions $(D_f^1, D_f^2, D_f^3, ...)$, all of which faithfully generate the output of $f$ over all points in $\mathcal{R}^d$, with the additional constraint that each $\mathcal{F}(D_f^i)$ should fit any $m$ datapoints sampled from $P^m$.

Given these observations and denominations, we then can define the Kolmogorov Growth of a function $f$ as follows:

$$KG_m(f) = \min_i \frac{\log \Pi_m \left( \mathcal{F}(D_f^i) \right)}{m}. \tag{2}$$

Note that $KG_m(f)$ requires the knowledge of distribution $P^m$ of the training samples. For an instance of the datapoints in $S$, we define the empirical Kolmogorov Growth via the empirical growth function $\widehat{\Pi}_S(\mathcal{F})$, which only computes the number of label assignments a function space can generate over the given $m$ training data samples in $S$. Thus, $\widehat{\Pi}_S(\mathcal{F}) = |\{(f(z_1), f(z_2), ..., f(z_m)) : f \in \mathcal{F}\}|$. This leads to the following definition of the empirical Kolmogorov Growth of a function $f$:

$$\widehat{KG}_S(f) = \min_i \frac{\log \widehat{\Pi}_S\left(\mathcal{F}(D_f^i)\right)}{m}. \tag{3}$$

**Remark:** Kolmorogov growth is indirectly motivated from Kolmogorov complexity itself. However, unlike Kolmogorov complexity, which is the length of the shortest program that generates $f$, Kolmogorov Growth is concerned with the *smallest function space* that $f$ can belong to, that can still fit the data well. Functions which have shorter descriptions usually require a smaller number of variables and are expected to have lower Kolmogorov Growth. Moreover, it turns out that Kolmogorov Growth allows us to directly comment on the error bounds for the function $f$ (see Section 3.1). We believe that a possible direction of future work would be to do a deeper study of the relationship between Kolmogorov Growth and Kolmogorov Complexity itself.

**Remark:** Note that, in the binary classification scenario, for a completely unstructured function $f$ (i.e., $f$ outputs random labels at every point $X \in \mathbb{R}^d$), one expects $KG(f)$ to be near its maximum value (i.e., $\log 2$). A structured $f$ would generate shorter descriptions with fewer parameters and therefore lead to smaller $KG(f)$.

### 3.1 Bounding Generalization Error using Kolmogorov Growth

Here we present error bounds that depend on $KG(f)$, where $f$ is the classification function, given the $m$ training data samples and their labels in $S$. As before, the data samples and labels in $S$ are drawn from some underlying distribution $P^m$.

We now define a set of error functions for computing training loss (0-1 loss) on $S$, denoted as $\widehat{err}_S(f)$, and the overall generalization error with respect to the distribution $P$, denoted by $err_P(f)$. We define them as follows:

$$\widehat{err}_S(f) = \sum_{i=1}^m \frac{(1 - f(z_i)y_i)}{2m} \tag{4}$$

$$err_P(f) = \mathbb{E}_{z,y \sim P}\left[\frac{(1 - f(z)y)}{2}\right]. \tag{5}$$

These definitions hold for any function $f$. Note that the error functions depend both on the function $f$ and the distribution $P$.

With this, we have the following results. The proofs of all results are provided in the supplementary material.

**Theorem 1** *For $0 < \delta < 1$, with probability $p \geq 1 - \delta$ over the draw of S, we have*

$$err_P(f) \leq \widehat{err}_S(f) + \sqrt{2KG_m(f)} + \sqrt{\frac{\log(1/\delta)}{2m}}. \tag{6}$$

The following corollary of the above theorem gives bounds that depend on empirical Kolmogorov growth $\widehat{KG}_S(f)$.

**Corollary 1.1** *For $0 < \delta < 1$, with probability $p \geq 1 - \delta$ over the draw of S, we have*

$$err_P(f) \leq \widehat{err}_S(f) + \sqrt{2\widehat{KG}_S(f)} + 4\sqrt{\frac{2\log(4/\delta)}{m}}. \tag{7}$$

**Remark:** Theorem 1 and its corollary essentially state that for functions $f$ of lower Kolmogorov growth, we should expect a smaller generalization gap. In what follows, we outline ways to approximate the empirical Kolmorogov growth $\widehat{KG}_S(f)$.

# 4 Teacher-Student Approximation Bounds for Kolmogorov Growth

The fundamental idea for approximating empirical Kolmogorov growth of a function $f$ which belongs to the function space $\mathcal{F}$ is to use a *student* classifier with a function space $\mathcal{F}^1_{small}$ (with a much smaller parametric count and complexity) to approximate the given function $f$ (the *teacher*). We apply this idea recursively. That is, if the function $f^1_{small} \in \mathcal{F}_{small}$ approximates $f$ best, we recursively estimate the empirical Kolmogorov growth of $f^1_{small}$ by approximating it via another classifier with a smaller function space $\mathcal{F}^2_{small}$ (thus, $\Pi_m(\mathcal{F}^2_{small}) < \Pi_m(\mathcal{F}^1_{small})$), and so on. We use this recursive way to then obtain a final estimate for $\widehat{KG}_S(f)$. We conjecture that, like Kolmogorov complexity itself, the true $\widehat{KG}_S(f)$ is uncomputable, so the estimate that results from this recursive approximation process is essentially an upper bound to the true $\widehat{KG}_S(f)$. The following theorem establishes an upper bound to empirical $KG$ approximation from a single smaller student classifier.

**Theorem 2** *Given the function $f \in \mathcal{F} : \mathbb{R}^d \to \mathbb{R}^2$ which outputs class logits for binary classification. We construct a function space $\mathcal{F}^1_{small}$ such that $\Pi_m(\mathcal{F}^1_{small}) < \Pi_m(\mathcal{F})$ and $\forall g \in \mathcal{F}^1_{small}$, there exists a description $D_g$ such that $\widehat{\Pi}_S(\mathcal{F}(D_g)) \leq \widehat{\Pi}_S(\mathcal{F}^1_{small})$. We approximate $f$ via another function $f^1_{small} \in \mathcal{F}^1_{small} : \mathbb{R}^d \to \mathbb{R}^2$ and let $\epsilon_{max}$ be such that*

$$\epsilon^2_{max}/2 = \max_{X \in \mathbb{R}^d} \|f^1_{small}(X) - f(X)\|^2. \tag{8}$$

*Denote the output probabilities generated from the corresponding logit outputs of $f(X)$ using the softmax operator (temperature $T = 1$), as $P_0(f(X))$ (label 1 output) and $P_1(f(X))$ (label 2 output). Let $0 \leq \delta \leq 1$ be such that*

$$Pr\left(\left|\log\left(\frac{P_0(f(X))}{P_1(f(X))}\right)\right| \leq \epsilon_{max}\right) \leq \delta, \tag{9}$$

*when $X$ is drawn from $S$. Then we have,*

$$\widehat{KG}_S(f) \leq \delta \log 2 + \frac{\log \widehat{\Pi}_S(\mathcal{F}^1_{small})}{m}, \tag{10}$$

*where $m$ is the number of samples in $S$.*

**Remark:** Theorem 2 demonstrates a way to bound the true empirical Kolmogorov growth of the function $f$, using a single student classifier function $f^1_{small} \in \mathcal{F}^1_{small}$. Note that unlike in Theorem 1, there are no direct constraints on the expressivity of $\mathcal{F}^1_{small}$, but rather a joint constraint on $\mathcal{F}^1_{small}$ and $\delta$ combined. If $\mathcal{F}^1_{small}$ cannot fit all $m$ points sampled from $P^m$, then the approximation error in $\delta$ will likely be higher, which will add to the estimate of $\widehat{KG}_S(f)$. The proof of Theorem 2 and its extension to the recursive approximation case are given in the supplementary material.

# 5 Network-to-Network (N2N) Regularization

We denote the base network to be trained as $N^{base}$ and the function modelled by the network weights $w^{base}$ as $N^{base}(w^{base}, X)$, where $X \in \mathbb{R}^d$ is the input. Here, $N^{base}(w^{base}, X)$ represents the output logits for the network $N^{base}$ when presented with the input $X$. Thus, we have $N^{base}(w^{base}, X) \in \mathbb{R}^c$, where $c$ is the number of classes. For what follows, let us denote the available training data and their labels by $S = \{X_i, y_i\}_{i=1}^m$.

The approach that follows is directly motivated from the result in Theorem 2. The main objective is to ensure that the *KG* of the network stays low during learning, using the teacher-student approximation error in Theorem 2. This is primarily achieved by ensuring that during training, the base network function $N^{base}(w^{base}, X)$ is always near to some function within the smaller network's function space. Next, we outline the details of the proposed multi-level network-to-network (N2N) regularization approach.

## 5.1 Multi-Level N2N: Details

In multi-level N2N regularization, we have multiple smaller networks $n^{small}_1, n^{small}_2, ..., n^{small}_K$ of decreasing complexity such that $\Pi_m(\mathcal{F}^1_{small}) > \Pi_m(\mathcal{F}^2_{small}) > \cdots > \Pi_m(\mathcal{F}^K_{small})$.

**Algorithm 1** N2N Regularization (Multi-Level)

---

**Input:** Training data $\{X_i, y_i\}_{i=1}^m$, base network $N^{base}$ and its weights $w^{base}$, $K$ networks $n_1^{small}, n_2^{small}, ..., n_K^{small}$, with weights $w_1, w_2, ..., w_K$ (s.t. $|n_1^{small}| > |n_2^{small}| > ..|n_K^{small}|$ in size), Number of epochs $J$, Hyperparameters $\lambda_0, \lambda_1, \lambda_2, ..., \lambda_{K-1}, \alpha_0, .., \alpha_K, e_{base}, e_{small}$.

1: **for** $j = 1, 2, \ldots, J$ **do**
2:     **for** $iter = 1, 2, \ldots, e_{base}$ **do**
3:         $\mathcal{L}_1 = \sum_{i=1}^m (L_{CE}(N^{base}(X_i), y_i) + \lambda_0 \|N^{base}(X_i) - n_1^{small}(X_i)\|^2)$
4:         Weight update: $w^{base} \leftarrow w^{base} - \frac{\alpha_0}{m}\frac{\partial \mathcal{L}_1}{\partial w^{base}}$
5:     **for** $k = 1, 2, \ldots, K$ **do**
6:         **for** $iter = 1, 2, \ldots, e_{small}$ **do**
7:             **if** $k = 1$ **then**
8:                 $\mathcal{L}_k = \sum_i \|N^{base}(X_i) - n_1^{small}(X_i)\|^2 + \lambda_1 \|n_2^{small}(X_i) - n_1^{small}(X_i)\|^2$
9:             **else if** $k = K$ **then**
10:                $\mathcal{L}_k = \sum_i \|n_k^{small}(X_i) - n_{k-1}^{small}(X_i)\|^2$
11:             **else**
12:                $\mathcal{L}_k = \sum_i \|n_k^{small}(X_i) - n_{k-1}^{small}(X_i)\|^2 + \lambda_k \|n_k^{small}(X_i) - n_{k+1}^{small}(X_i)\|^2$
13:         Weight update: $w_k \leftarrow w_k - \frac{\alpha_k}{m}\frac{\partial \mathcal{L}_k}{\partial w_1}$

---

The corresponding functions resulting from the network weights $w_1, w_2, .., w_K$ are denoted as $n_1^{small}(w_1, X), n_2^{small}(w_2, X), ..., n_K^{small}(w_K, X)$. Next, we outline the loss functions for all networks. For the larger to-be-trained base network $N^{base}$, the loss objective is to minimize cross-entropy loss on $S$ while being close to $n_1^{small}(w_1, X)$ for some choice of weights $w_1$ ($\mathcal{L}_1$ in Algorithm 1). For the smaller network $n_1^{small}$, the objective is two-fold: find the weight configuration $w_1$ that approximates the larger network function $N^{base}$, while also being close to $n_2^{small}(w_2, X)$ for some choice of $w_2$ ($\mathcal{L}_2$ in Algorithm 1). Thus, we force the smaller network $n_1^{small}$ to be close to the base network and an even lower-complexity network $n_2^{small}$ at the same time. Similarly we can define $\mathcal{L}_3, \mathcal{L}_3, .., \mathcal{L}_{K-1}$, except for $\mathcal{L}_K$ which applies to the smallest network $n_K^{small}$. The loss objective for $n_K^{small}$ is to just keep $n_K^{small}(w_K, X)$ close to $n_{K-1}^{small}(w_{K-1}, X)$. Finally, we optimize the loss functions in an alternating manner in the order of $\mathcal{L}_1, \mathcal{L}_2, .., \mathcal{L}_K$. Details are given in Algorithm 1. The choice of mean-squared error based loss functions here directly follows from the result in Theorem 2. Note that Algorithm 1 updates with the entire batch of training data points at each iteration, and can be extended to the case of minibatch stochastic gradient descent (SGD).

### 5.2 Other Relevant Approaches in Literature

To the best of our knowledge, our proposed approach is novel, and we did not find much relevant work. Conceptually, we found the reverse knowledge distillation method [16] to be the most relevant to our proposed approach, which regularizes large teacher networks using smaller, trained versions of student networks of less depth. The outputs logits of the trained student networks are then essentially re-used for *smoothing* the outputs of the larger neural network. Here, we do not directly use trained student networks to supervise the teacher, but instead simply ensure that during training, the teacher network is within reach of *some* student network (which may change throughout the training process), which is a more relaxed constraint. Also, the mean-squared error based approximation error between the student and teacher networks is motivated from Theorem 2, and differs from KL-divergence based measures used in knowledge distillation. Another point of difference is that N2N uses a multi-level approach for a recursive way of regularizing multiple networks of different levels of complexity.

## 6 Experiments

We test N2N on three datasets: MNIST [17], CIFAR-10 [18] and CIFAR-100 [19]. We also demonstrate that N2N regularization improves performance in the presence of label noise. Lastly, we analyse Kolmogorov growth of networks during training. Experiments were either carried out on an RTX 2060 GPU or a Tesla V100 or A100 GPU. As mentioned in Algorithm 1, an epoch refers to a total of $e_{base}$ iterations of training the base network and $e_{small}$ iterations of training the smaller networks on the whole dataset. Code will be made available at https://github.com/rghosh92/N2N.

| | MNIST | | | | CIFAR-10 | | | | CIFAR-100 | | | |
|---|---|---|---|---|---|---|---|---|---|---|---|---|
| Data Size (Training) | no reg | drop+l2 | drop+l2 +N2N-1 | drop + l2 + N2N-2 | no reg | l2-norm | l2 +N2N-1 | l2 + N2N-2 | no reg | l2-norm | l2 +N2N-1 | l2 + N2N-2 |
| 1000 | $96.74_{\pm0.11}$ | $97.2_{\pm0.05}$ | $97.42_{\pm0.12}$ | $\mathbf{97.62}_{\pm0.08}$ | $45.35_{\pm1.25}$ | $48.99_{\pm0.71}$ | $50.58_{\pm0.65}$ | $\mathbf{51.19}_{\pm0.44}$ | $6.21_{\pm0.18}$ | $7.97_{\pm0.12}$ | $8.96_{\pm0.23}$ | $\mathbf{9.45}_{\pm0.2}$ |
| 2000 | $97.71_{\pm0.09}$ | $98.06_{\pm0.03}$ | $98.16_{\pm0.07}$ | $\mathbf{98.27}_{\pm0.03}$ | $51.88_{\pm0.47}$ | $55.42_{\pm0.4}$ | $56.24_{\pm0.85}$ | $\mathbf{57.23}_{\pm0.7}$ | $8.84_{\pm0.15}$ | $9.75_{\pm0.1}$ | $13.01_{\pm0.2}$ | $\mathbf{13.75}_{\pm0.16}$ |
| 10000 | $98.92_{\pm0.04}$ | $99_{\pm0.01}$ | $99.07_{\pm0.01}$ | $\mathbf{99.13}_{\pm0.01}$ | $71.84_{\pm0.29}$ | $72.13_{\pm0.29}$ | $72.55_{\pm0.31}$ | $\mathbf{72.89}_{\pm0.38}$ | $33.51_{\pm0.09}$ | $35.03_{\pm0.05}$ | $35.57_{\pm0.12}$ | $\mathbf{35.73}_{\pm0.04}$ |
| Complete | $99.38_{\pm0.04}$ | $99.59_{\pm0.03}$ | $99.68_{\pm0.04}$ | $\mathbf{99.70}_{\pm0.02}$ | $92.53_{\pm0.07}$ | $92.96_{\pm0.04}$ | $\mathbf{93.35}_{\pm0.05}$ | $93.26_{\pm0.09}$ | $75.65_{\pm0.1}$ | $76.32_{\pm0.07}$ | $76.65_{\pm0.1}$ | $\mathbf{76.83}_{\pm0.02}$ |

Table 1: Test Accuracy on MNIST, CIFAR-10, and CIFAR-100 for different training data sizes and different regularization choices (dropout, L2-norm, single-level N2N, multi-level N2N and their combinations). We note that N2N combined with dropout and L2-norm or only L2-norm leads to improvements in test accuracy for all cases. This shows that N2N regularization complements other regularization approaches well, improving generalization performance of trained networks further.

## 6.1 Supervised Classification: MNIST, CIFAR-10, CIFAR-100

The primary objective of the experiments presented here is to see whether N2N regularization can drive the training process towards network configurations that generalize better. For each dataset, results are reported for various choices of training data size. Furthermore, to show that our regularization approach complements other commonly used regularization approaches, we show results when our approach is combined with Dropout and L2-norm regularization. For the ResNet networks (CIFAR-10/100), we combine N2N with L2-norm regularization. All networks were trained for a total of 200 iterations,

| $\underset{m\in(1000,2000)}{\mathbb{E}}\left[\frac{1}{\log 2}\left(\widehat{KG}_S(f) - \frac{\log\widehat{\Pi}_S\left(\mathcal{F}^1_{small}\right)}{m}\right)\right]$ | | | |
|---|---|---|---|
| | no reg | drop + l2 | drop+l2+ N2N-2 |
| MNIST | 0.0046 | 0.0032 | 0.0015 |
| CIFAR-10 | 0.4721 | 0.4545 | 0.4427 |
| CIFAR-100 | 0.725 | 0.6901 | 0.6324 |

Table 2: Averaged approximation error term ($\delta$ in Theorem 2) for the networks trained on MNIST, CIFAR-10 and CIFAR-100 datasets for $m = 1000, 2000$. Smaller values of $\delta$ imply lower $\widehat{KG}_S(f)$ of the trained networks.

and in each case results reported are averaged over five networks. For all experiments we set $e_{base} = 3$, $e_{small} = 1$ in Algorithm 1. The values of the regularization parameters $(\lambda_0, \lambda_1)$ are provided in the supplementary material. Note that due to the additional iterations for training the smaller networks, the worst case training time for the N2N approach is 1.5 times longer than standard training. Across all three datasets, we generally find that for larger training data sizes, smaller regularization parameters yield best performance, reinforcing the fact that N2N is indeed a form of regularization. This is primarily because for large training data, the distribution is dense enough for the network to learn, and thus less emphasis can be given to the N2N regularization term.

Results are shown in Table 1, and the average approximation error $\delta$ for the trained networks is shown in Table 2. We note that the use of N2N regularization improves test accuracy. Mainly, we see that N2N regularization complements common regularization approaches such as dropout and L2-norm well. In all cases we find that combining these well-known regularization approaches with the proposed approach yields the best results. Furthermore, we also see that the improvement in performance persists when the training data size is increased. Lastly, in most cases, we see that 2-level N2N regularization (N2N-2, $m = 2$ in Algorithm 1), outperforms single-level N2N (N2N-1, $m = 1$ in Algorithm 1), with the exception of CIFAR-10 with the full training dataset. For the CIFAR-10 and CIFAR-100 datasets, we used the benchmark ResNet architectures ResNet-44 and ResNet-50 respectively. Our results with L2-norm regularization for the ResNet-44 and ResNet-50 architectures are slightly better than the results originally reported in [20]. For the MNIST dataset, we used a 5-layer CNN with 3 conv layers and 2 fc layers. Network architecture details are provided in the supplementary material. Note that although better results can be found in literature, our objective was to demonstrate that using N2N regularization in conjunction with common regularization approaches can benefit both shallow CNN architectures (MNIST) and ResNets (CIFAR-10, CIFAR-100). Furthermore, as Table 2 shows, we find that N2N reduces the empirical KG of trained networks, and datasets on which test accuracies are lower yield higher KG of trained networks. This supports the implications of Theorem 1, as high KG functions are expected to have a larger generalization gap.

## 6.2 Learning with Noisy Labels

As our proposed regularization approach constrains the network function to be simpler by minimizing an approximation of Kolmogorov Growth, it naturally applies to the case of noisy training labels.

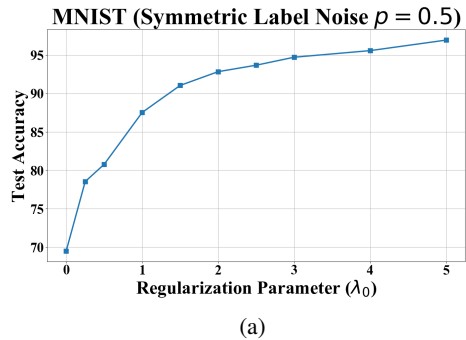
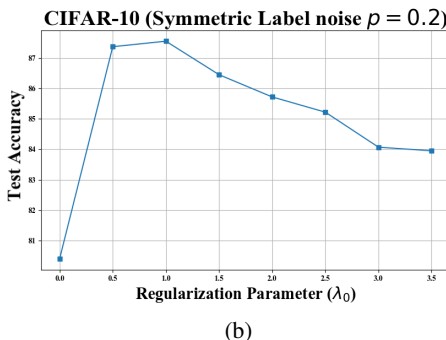

| (a) | (b) |

Figure 1: The effect of the hyperparameter $\lambda_0$ of our proposed N2N regularization approach on test accuracy on the MNIST (symmetric noise $p = 0.5$) and CIFAR-10 (symmetric noise $p = 0.2$) datasets. The networks were regularized using 2-level N2N, with a fixed $\lambda_1 = 0.1$. In both datasets, we see significant performance improvements in response to regularization.

| Noise level | F-Correction [22] | Decoupling [23] | MentorNet [24] | Co-Teaching [25] | SCE [21] | N2N (2-Levels) |
|---|---|---|---|---|---|---|
| | | | MNIST | | | |
| $p = 0.2$ | $98.80_{\pm0.12}$ (94.05) | $95.70_{\pm0.02}$ (94.05) | $96.70_{\pm0.22}$ (94.05) | $97.25_{\pm0.03}$ (94.05) | $\mathbf{99.30}_{\pm0.02}$ (93.57) | $99.09_{\pm0.6}$ (93.57) |
| $p = 0.5$ | $79.61_{\pm1.96}$ (66.05) | $81.15_{\pm0.03}$ (66.05) | $90.05_{\pm0.30}$ (66.05) | $91.32_{\pm0.06}$ (66.05) | $97.87_{\pm0.27}$ (65.63) | $\mathbf{98.32}_{\pm0.05}$ (65.63) |
| | | | CIFAR-10 | | | |
| $p = 0.2$ | $84.55_{\pm0.16}$ (76.25) | $80.44_{\pm0.05}$ (76.25) | $80.76_{\pm0.36}$ (76.25) | $82.32_{\pm0.07}$ (76.25) | $\mathbf{88.91}_{\pm0.04}$ (86.42) | $88.06_{\pm0.06}$ (86.42) |
| $p = 0.5$ | $59.83_{\pm0.17}$ (48.87) | $51.49_{\pm0.08}$ (48.87) | $71.10_{\pm0.48}$ (48.87) | $74.02_{\pm0.48}$ (48.87) | $\mathbf{83.91}_{\pm0.02}$(77.15) | $80.95_{\pm0.19}$ (77.15) |
| | | | CIFAR-100 | | | |
| $p = 0.2$ | $61.87_{\pm0.21}$ (47.55) | $44.52_{\pm0.04}$ (47.55) | $52.13_{\pm0.40}$ (47.55) | $54.23_{\pm0.08}$ (47.55) | $65.96_{\pm0.83}$ (64.4) | $\mathbf{66.83}_{\pm0.77}$ (64.4) |
| $p = 0.5$ | $41.04_{\pm0.07}$ (25.21) | $25.80_{\pm0.04}$ (25.21) | $39.00_{\pm1.00}$ (25.21) | $41.37_{\pm0.08}$ (25.21) | $49.27_{\pm0.44}$ (48.74) | $\mathbf{54.79}_{\pm1.31}$ (48.74) |

Table 3: Test performance on MNIST, CIFAR-10 and CIFAR-100 when symmetric label noise of probabilities $p = 0.5$ and $p = 0.2$ was applied on the training data labels. Average accuracies are reported for various state-of-the-art and benchmark approaches that have been recently proposed in literature, including benchmark approaches such as symmetric cross entropy (SCE [21]). The numbers within the brackets represent the test accuracy of the corresponding network architecture when trained with standard cross-entropy loss, and thus the relative reduction in error can be construed as a measure of effectiveness of each approach. We note that N2N regularization helps improve on the cross-entropy baseline in all cases, significant improving in some of them.

Without regularization, label noise in the training data usually forces a network to emulate a more *complex* function, as it potentially makes the decision boundary more complex, a fact that we also empirically observe in section 6.3. We stipulate that N2N regularization should help the network in achieving simpler functions to approximate the training data labels, favoring simpler decision boundaries over complex ones, and thus potentially shielding against the corrupted labels to a certain extent. We test whether enforcing a simpler function (large $\lambda_0, ...\lambda_m$) at the cost of compromising training loss can help improve test accuracy, when the training data is corrupted by label noise. We tested the cases where symmetric and asymmetric label noise of some probability $p$ was applied (same as in [21]), and show our results for symmetric noise with $p = 0.5$ and $p = 0.2$. Results with asymmetric pair-flip noise of probability $p = 0.45$ are shown in the Supplementary Material.

First, we show the results for symmetric label noise of probability $p = 0.5$ and $p = 0.2$ on MNIST, CIFAR-10 and CIFAR-100 in Table 3. For F-correction [22], Decoupling [23], MentorNet [24] and Co-Teaching [25] methods, we report the accuracy over the last ten iterations of training as observed in [25], along with their standard cross-entropy results with corresponding network architectures for reference. We do the same for our implemented SCE and N2N methods on MNIST and CIFAR-10, but for CIFAR-100, we report the accuracies using a 48k-2k training-validation split of the data for both, as we find it to yield best performance (due to hard convergence). Note that we use the same network configurations for SCE. The values of $\lambda_0$ and $\lambda_1$ are provided in the supplementary material. We find that N2N regularization yields competitive performance in most cases. We also plot the test accuracy as a function of the regularization parameter $\lambda_0$ in Figure 1. We find that for MNIST, large $\lambda_0$ helps achieve significantly higher test accuracy, whereas for CIFAR-10 and CIFAR-100 accuracy peaks around $\lambda_0 = 0.6$ and $\lambda_0 = 0.5$ respectively. Note that accuracies may differ from Table 3 because of different training configurations used.

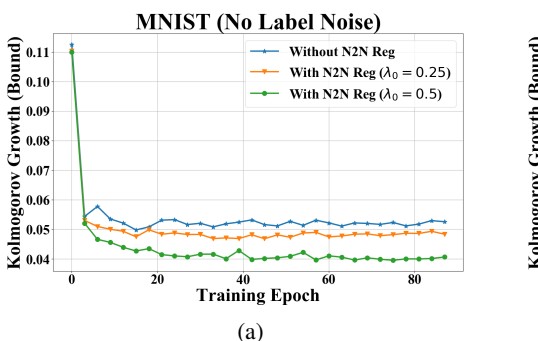
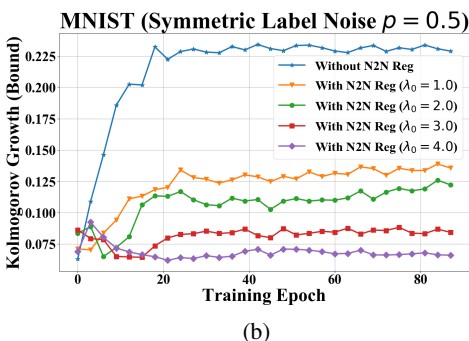

(a)                            (b)

Figure 2: We show the trend of the $KG$ of networks (approximated using Theorem 2) during the training process on MNIST-1k (with and without label noise). In (a), we plot how the $KG$ of networks changes during training on MNIST (no label noise). There, we compare the trend for three different cases: no N2N regularization, N2N regularization with $\lambda_0 = 0.25$, and N2N regularization with $\lambda_0 = 0.50$. As $\lambda_0$ increases, we see that the $KG$ of the trained networks show a clear decrease, with the trajectories deviating after a few epochs of training. In (b), we show $KG$ of networks during training on the MNIST dataset corrupted with symmetric label noise of probability $p = 0.5$. There, we find that the networks trained without any N2N regularization have significantly higher $KG$ than the networks trained with N2N regularization. Particularly, we find that as $\lambda_0$ increases, $KG$ reduces, while also improving test performance (see Fig. 1). All values of $KG$ estimated here are 10-fold averages over multiple training sessions with randomly initialized networks.

## 6.3 Comparing Kolmogorov Growth Trajectories during Training

The improvements observed via the use of N2N regularization lead to the question of how the network trajectories differ when N2N regularization is used, as compared to when it is not applied. We use the result in Theorem 2 to compute the bounded approximation to empirical Kolmogorov growth of the network function. We thus plot the approximation of $\widehat{KG}(f)$ of the function $f$ represented by the neural network during the training process. Note that the variation of KG in all plots is only owing to the changes in the approximation error term $\delta$ in Theorem 2, as $\mathcal{F}^1_{small}$ is fixed to a single-layer CNN (of a fixed configuration) for all results in Fig. 2. $\widehat{\Pi}_S\left(\mathcal{F}^1_{small}\right)$ was estimated using a VC dimension based approximation shown in [7]. Results are shown in Figure 2. We find that in the case of no training label noise, the $KG$ of networks typically have high initial values, steeply reducing within a few epochs of training, after which it stabilizes. Expectedly, when trained with N2N regularization, we find that the final $KG$ of networks are lower, compared to $KG$ of networks trained without N2N. In the case of label noise, we report some interesting observations. First, we see that, differently from before, the $KG$ values rather increase with training and stabilize eventually at higher values. This is almost an opposite trend to the case of no label noise. This can be partly explained by the fact that as training progresses, the network slowly adapts its decision boundary to fit the erroneous labelling, eventually resulting in a decision boundary of high complexity. For the label noise case, we find that N2N regularization significantly reduces the increase of $KG$ during the training process. Furthermore, larger values of the $\lambda$ parameter leads to networks which exhibit smaller $KG$ values. This also helps explain the significant gains in test accuracy observed for MNIST earlier in Table 3, when using N2N regularization.

## 7 Discussion and Reflections

This results in this paper further the recent work by [5], where it was shown that neural networks are inherently biased towards *simpler* functions of lower Kolmogorov complexity. In particular, we provide an actionable method for incorporating a function complexity prior while learning, using a novel measure called Kolmogorov Growth. Unlike Kolmogorov complexity, which is the description of the shortest program that generates some function $f$, Kolmogorov Growth is concerned with the *smallest function space* that $f$ can belong to, that can still fit the data well. Functions with shorter descriptions will typically need fewer variables and thus may have lower Kolmogorov Growth. Although smaller function spaces have less expressive power, as recent work in [6] shows, even

shallower neural nets can fit random labels on the training data points. The observations in [5] however, put a new perspective on the results in [6]: any *random* choice of network weights on smaller networks is likely to yield a low complexity function. Thus, even shallower networks can potentially exhibit a wide range of complexities. Among them, the higher Kolmogorov complexity functions are likely required for a network to fit random labels (similar to observations in [5]). In the case of label noise, N2N considers this fact by avoiding directly training the shallower networks to fit the noisy labels, which helps reduce their descriptional complexity, which then helps in regularizing the larger base network. As such, when using a pre-trained shallower network to regularize the base network (reverse-KD), we found that performance can significantly suffer in the case of label noise.

Via N2N regularization we see that enforcing low $KG$ for large networks can improve their ability to generalize. The proposed approach greatly helps in the scenario where the training data has noisy labels, attaining competitive performance on the three tested datasets when the training labels are corrupted with symmetric label noise. In the case of label noise, we see that networks trained without N2N regularization have larger Kolmogorov Growth (see Figure 2), which reduces immediately following the application of N2N regularization. Furthermore, it is clear that by varying the emphasis on minimizing the regularization term via tuning the $\lambda$ parameters, the $KG$ of subsequently trained networks can effectively be controlled. As $\lambda_0$ increases, more emphasis is put on lowering KG, which improves generalization and yields better test accuracy. However, this happens up to a threshold (see Figure 1) and test accuracy decreases as $\lambda_0$ increases beyond the threshold. In the case of training data with noisy labels, we find that the threshold is larger because we can put less emphasis on fitting the noisy training labels and more emphasis on minimizing KG.

Our theoretical results in Section 3.1 show that network configurations that can be approximated well by smaller networks of lower complexity will have low Kolmogorov growth, and subsequently, lower generalization error. These results concur with the very recent theoretical findings in [26], which finds analogous results for the Rademacher complexity based generalization error framework, in the context of knowledge distillation. Our main result in Theorem 1 outlines an Occam's razor like principle for generalization. Theorem 1 implies that for all functions which have zero training error, the function with the smallest $KG_m(f)$ will be the most likely to show the least generalization error.

Our empirical findings consistently show that driving the networks towards simpler functions of lower Kolmogorov Growth leads to networks that generalize better. We find multi-level N2N follows from a theoretical result shown in the Supplementary material, where we bound the empirical KG of the base network function based on the set of recursive mean-squared error estimates. However, KG bounds resulting from recursive estimation are provably less tight than single-estimation KG bounds of the form in Theorem 2. We believe that more bounded loss terms could be one of the reasons behind 2-level N2N yielding better performance on average, as compared to single-level N2N.

In the case of label noise, we see that enforcing low $KG_m(f)$ on the classification function $f$, by increasing the regularization parameter values, can have a significant impact (Section 6.2). This also leads to a current limitation of our approach, which is that the hyperparameters $(\lambda_0, \lambda_1, ..)$ have to be manually tuned. Automatic estimation of their optimal values is an avenue for future research. Another limitation of our work is that the growth function term in the empirical approximation of KG (Theorem 2) potentially can render the bounds quite loose. Thus, achieving tighter bounds with KG-based metrics is also a possible extension of this work.

In N2N regularization, we observe that the properties of the smaller networks can dictate the learning of the base network. If we choose smaller networks which are highly rotation invariant in their structure (for e.g., by using a rotation-invariant CNN), we should expect the base network to adopt some of the rotation invariance properties as well. We thus conducted an additional experiment on a custom MNIST [17] dataset, which contains images of digits translated randomly within the image. We added symmetric noise on the labels ($p = 0.5$), and tested our proposed N2N regularization approach with a student network which is highly translation invariant (large max-pooling windows). We found that N2N shows larger improvements, reducing test error by 27% compared to other baselines. This demonstrates the possibility of extending this work by analyzing the effect of invariance/equivariance choices in the smaller networks on the generalization behaviour of the larger network, similar to the observations on distillation methods transferring inductive biases in [27].

Finally, since our work provides a certain level of robustness against label noise, it supports activities such as crowdsourcing data labelling, which potentially contains significant label noise.

## 8  Acknowledgements

This research was supported by the National University of Singapore and by A*STAR, CISCO Systems (USA) Pte. Ltd and National University of Singapore under its Cisco-NUS Accelerated Digital Economy Corporate Laboratory (Award I21001E0002). We would also like to acknowledge the helpful feedback provided by members of the Kent-Ridge AI research group at the National University of Singapore.

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
