# Supplementary Material: Network-to-Network Regularization: Enforcing Occam's Razor to Improve Generalization

**Rohan Ghosh and Mehul Motani**
Department of Electrical and Computer Engineering
N.1 Institute for Health
Institute of Data Science
National University of Singapore
rghosh92@gmail.com, motani@nus.edu.sg

## 1  Summary of Supplementary Material

In this supplementary material, we first provide more details regarding the experiments conducted in the main paper, after which we report some additional results on MNIST and CIFAR-10 in the asymmetric label noise setting. Finally, we provide the proofs of all theoretical results.

## 2  Experimental Details

### 2.1  Regularization Parameters

Tables 1 and 2 provide the details of the regularization parameters $\lambda_0, \lambda_1$ used in the proposed N2N algorithm, for the main experiment (Section 6.1) and the label-noise experiment (Section 6.2) of the main paper. We usually see that increasing $\lambda_0$ in steps to its final value yields better convergence.

| Training Data Size | MNIST | | CIFAR-10 | | CIFAR-100 | |
|---|---|---|---|---|---|---|
| | $\lambda_0$ | $\lambda_1$ | $\lambda_0$ | $\lambda_1$ | $\lambda_0$ | $\lambda_1$ |
| 1000 | 0.5 | 0.5 | 0.1 | 0.5 | 0.1 | 0.5 |
| 2000 | 0.25 | 0.25 | 0.1 | 0.1 | 0.1 | 0.1 |
| 10000 | 0.25 | 0.1 | 0.05 | 0.1 | 0.02 | 0.1 |
| Complete Dataset | 0.2 | 0.1 | 0.01 | 0.1 | 0.01 | 0.1 |

Table 1: Values assigned to the regularization parameters $\lambda_0$ and $\lambda_1$ for the results shown in Table 1 of the main paper. For the single-level N2N, we simply set $\lambda_1 = 0$.

| Label Noise Probability | MNIST | | CIFAR-10 | | CIFAR-100 | |
|---|---|---|---|---|---|---|
| | $\lambda_0$ | $\lambda_1$ | $\lambda_0$ | $\lambda_1$ | $\lambda_0$ | $\lambda_1$ |
| p=0.2 | 4.0 | 0.1 | 0.6 | 0.5 | 0.3 | 0.1 |
| p=0.5 | 10 | 0.1 | 0.6 | 0.5 | 0.5 | 0.1 |

Table 2: Values assigned to the regularization parameters $\lambda_0$ and $\lambda_1$ for the results shown in Table 2 of the main paper (MNIST and CIFAR-10) and Table 1 of this supplementary material (CIFAR-100). $p$ represents the symmetric noise probability used for generating the corrupted labels.

### 2.2  Network Architecture Details

We report the network architecture details and their parameter counts for the networks used in the proposed N2N regularization approach across the MNIST, CIFAR-10 and CIFAR-100 datasets. We note that for 2-Level N2N regularization, we would have a total of three networks in decreasing order of complexity; the base network $N^{base}$, the level-1 network $n^1_{small}$, and the level-2 network $n^2_{small}$.

35th Conference on Neural Information Processing Systems (NeurIPS 2021).

| Network | MNIST | CIFAR-10 | CIFAR-100 |
|---|---|---|---|
| $N^{base}$ | 3 Conv + 2 FC (577K) | ResNet-44 (658K) | ResNet-50 (23M) |
| $n^1_{small}$ | 2 Conv + 2 FC(25K) | 2 Conv + 3 FC (28K) | 2 Conv + 3 FC (34K) |
| $n^2_{small}$ | 1 Conv + 2 FC (5K) | 1 Conv + 3 FC (11K) | 1 Conv + 3 FC (16K) |

Table 3: Network architectures and parameter counts for networks used in MNIST, CIFAR-10 and CIFAR-100 datasets, for each level of the N2N regularization algorithm. Note that Conv denotes convolutional layers and FC denotes fully connected layers.

| Noise level | F-Correction [1] | Decoupling [2] | MentorNet [3] | Co-Teaching [4] | SCE [5] | N2N (2-Levels) |
|---|---|---|---|---|---|---|
| | | | MNIST, Asymmetric Pairwise Noise | | | |
| $p = 0.45$ | $0.24_{\pm 0.03}$ (56.52) | $58.03_{\pm 0.07}$ (56.52) | $80.88_{\pm 4.45}$ (56.52) | $87.63_{\pm 0.21}$ (56.52) | $90.785_{\pm 0.51}$ (55.58) | $\mathbf{91.08}_{\pm 0.83}$ (55.58) |
| | | | CIFAR-10, Asymmetric Pairwise Noise | | | |
| $p = 0.45$ | $6.61_{\pm 0.03}$ (49.5) | $48.80_{\pm 0.04}$ (49.5) | $58.14_{\pm 0.38}$ (49.5) | $72.62_{\pm 0.15}$ (49.5) | $78.71_{\pm 0.08}$ (74.17) | $\mathbf{81.38}_{\pm 0.19}$ (74.17) |

Table 4: Test accuracies on MNIST and CIFAR-10 when pairflip label noise of probability $p = 0.45$ was applied on the training data labels. Average accuracies over last ten epochs and standard cross-entropy accuracies with the corresponding network configurations (in brackets) are shown here for various benchmark approaches as observed in [4], and the SCE results here are computed with our network and training configurations.

Please refer to Algorithm 1 of the main paper for more details. The network architectures for all levels and their parameter counts are provided in Table 3.

## 2.3 Computing Kolmogorov Growth on MNIST

For estimating $\widehat{KG}_S(f)$ (where $S$ has $m$ datapoints) we use $n^2_{small}$ to bound the empirical $KG$ of $N^{base}$, by finding the configuration of $n^2_{small}$ which minimizes the mean-squared error between $n^2_{small}$ and $N^{base}$ on data separate from $S$ but also generated by $P$. And then we compute empirical KG based on the average label disagreement ($\delta$) between the resulting $n^2_{small}$ and $N^{base}$ on $S$.

## 3 Additional Experiments: Asymmetric Label Noise

We report the results of applying the proposed N2N regularization approach in the case of MNIST and CIFAR-10 with asymmetric pairwise label noise of probability 0.45 (same as in [4]). Results are shown in Table 4. We trained the N2N and SCE networks for 100 iterations with the same training configuration, and report the average accuracy over the last 10 iterations. Also, in Figure 1, we

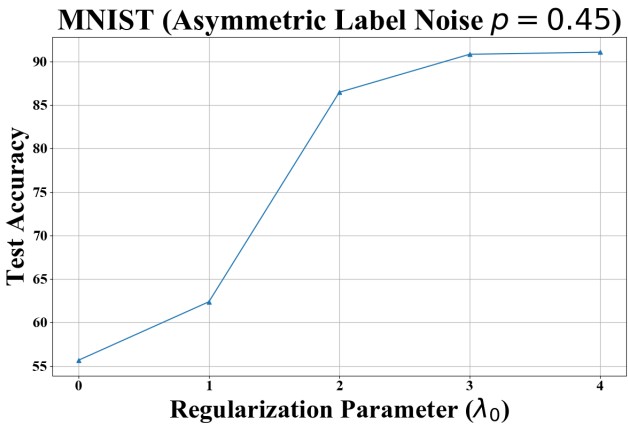

Figure 1: Average test accuracy over the last ten iterations of training with 2-level N2N, as a function of the regularization parameter $\lambda_0$, in the case of MNIST, with pairwise label noise of 45%.

demonstrate the impact of the regularization parameter $\lambda_0$ in the case of MNIST with asymmetric noise of $p = 0.45$. We see that similar to the case of symmetric noise, increasing $\lambda_0$ helps in significantly improving the network's ability to generalize.

# 4 Proofs of Theoretical Results

For all results that follow, we consider the case of binary classification. Also, please refer to Section 3 of the main paper for all definitions. In addition to the theoretical results mentioned in the main paper, as referenced in Section 4 of the main paper, we provide an additional result for recursive teacher-student approximation in Corollary 2.1.

## 4.1 Proof of Theorem 1 and Corollary 1.1

**Theorem 1.** *For $0 < \delta < 1$, with probability $p \geq 1 - \delta$ over the draw of S, we have,*

$$err_P(f) \leq \widehat{err}_S(f) + \sqrt{2KG_m(f)} + \sqrt{\frac{\log{(1/\delta)}}{2m}}. \tag{1}$$

*Proof.* First, we re-iterate the well known result [6] that bounds generalization error based on the Rademacher complexity for a function space $\mathcal{F}$, denoted as $\mathcal{R}_m(\mathcal{F})$. It states that for $0 < \delta < 1$, with probability $p \geq 1 - \delta$ over the draw of S, we have for any function $f' \in \mathcal{F}$,

$$\text{err}_P(f') \leq \widehat{\text{err}}_S(f') + \mathcal{R}_m(\mathcal{F}) + \sqrt{\frac{\log{(1/\delta)}}{2m}}. \tag{2}$$

Next, it is well known that the application of Massart's finite class lemma allows us to bound the Rademacher complexity in terms of the growth function as $\mathcal{R}_m(\mathcal{F}) \leq \sqrt{\frac{2\log{\Pi_m(\mathcal{F})}}{m}}$.

Thus we can re-iterate the result in (2) as follows.

For $0 < \delta < 1$, with probability $p \geq 1 - \delta$ over the draw of S, we have for any function $f' \in \mathcal{F}$,

$$\text{err}_P(f') \leq \widehat{\text{err}}_S(f') + \sqrt{\frac{2\log{\Pi_m(\mathcal{F})}}{m}} + \sqrt{\frac{\log{(1/\delta)}}{2m}}. \tag{3}$$

Next, in the context of the computation of the Kolmogorov Growth $KG_m(f)$ of a function, we define the *minimal* function space $M_f$ corresponding to a function $f$, as the function space resulting from the description $\mathcal{D}_f^j$, where $\mathcal{D}_f^j$ is chosen such that,

$$j = \operatorname*{argmin}_i \frac{\log{\Pi_m\left(\mathcal{F}(D_f^i)\right)}}{m}. \tag{4}$$

Given the above formulation, we have that $M_f = \mathcal{F}(\mathcal{D}_f^j)$. With this, we note that the error bound based on the growth function should apply to any function space, including $M_f$, as follows. For $0 < \delta < 1$, with probability $p \geq 1 - \delta$ over the draw of S, we have for any function $g \in M_f$,

$$\text{err}_P(g) \leq \widehat{\text{err}}_S(g) + \sqrt{\frac{2\log{\Pi_m(M_f)}}{m}} + \sqrt{\frac{\log{(1/\delta)}}{2m}}. \tag{5}$$

Thus, we note that for a random draw of S, according to its i.i.d probability $P(S)$, the above holds. Now, we make use of the fact that $M_f$ can fit any set of $m$ sampled points from $P(S)$ (due to the constraints described in Section 3 of the main paper). This indicates that if $M_f$ can be triggered from any set of $m$ sampled points, i.e., knowledge of $M_f$ does not give any additional information about the set of $m$ sampled points in the training dataset $S$. This then indicates that $P(S|M_f) = P(S)$, and the result in (5) holds for the scenario where $g = f$, leading to the following.

For $0 < \delta < 1$, with probability $p \geq 1 - \delta$ over the draw of S, we have for any function $f$,

$$\text{err}_P(f) \leq \widehat{\text{err}}_S(f) + \sqrt{\frac{2\log{\Pi_m(M_f)}}{m}} + \sqrt{\frac{\log{(1/\delta)}}{2m}}. \tag{6}$$

The main result then follows by simply noting that by its definition, $KG_m(f) = \frac{\log \Pi_m(M_f)}{m}$. $\quad\square$

**Corollary 1.1.** *For $0 < \delta < 1$, with probability $p \geq 1 - \delta$ over the draw of $S$, we have*

$$err_P(f) \leq \widehat{err}_S(f) + \sqrt{2\widehat{KG}_S(f)} + 4\sqrt{\frac{2\log(4/\delta)}{m}}. \tag{7}$$

*Proof.* To prove the above, we first note the error bounds based on empirical Rademacher complexity, as proposed in [6] as follows. It states that for $0 < \delta < 1$, with probability $p \geq 1 - \delta$ over the draw of $S$, we have for any function $f' \in \mathcal{F}$,

$$\mathrm{err}_P(f') \leq \widehat{\mathrm{err}}_S(f') + \widehat{\mathcal{R}}_S(\mathcal{F}) + 4\sqrt{\frac{2\log(4/\delta)}{m}}. \tag{8}$$

Next, we use the result from Massart's finite class lemma, which bounds the empirical Rademacher complexity in terms of the empirical growth function $\widehat{\Pi}_S(\mathcal{F})$ to change the statement in (8) as follows. For $0 < \delta < 1$, with probability $p \geq 1 - \delta$ over the draw of $S$, we have for any function $f' \in \mathcal{F}$,

$$\mathrm{err}_P(f') \leq \widehat{\mathrm{err}}_S(f') + \sqrt{\frac{2\log\widehat{\Pi}_S(\mathcal{F})}{m}} + 4\sqrt{\frac{2\log(4/\delta)}{m}}. \tag{9}$$

Next, in the context of the computation of empirical Kolmogorov Growth $\widehat{KG}_S(f)$ of the given function $f$, we define the empirical *minimal* function space $\widehat{M}_S(f)$ corresponding to a function $f$ and an instance of the training data in $S$, as the function space resulting from the description $\mathcal{D}_f^j$, where $\mathcal{D}_f^j$ is chosen such that,

$$j = \underset{i}{\mathrm{argmin}} \; \frac{\log\widehat{\Pi}_S\left(\mathcal{F}(D_f^i)\right)}{m}. \tag{10}$$

Given the above formulation, we have that $\widehat{M}_S(f) = \mathcal{F}(\mathcal{D}_f^j)$. Using this, one could construct the error bounds in the following manner, by constraining $f$ to belong to the function space $\widehat{M}_S(f)$. For $0 < \delta < 1$, with probability $p \geq 1 - \delta$ over the draw of $S$, we have for any function $g \in \widehat{M}_S(f)$,

$$\mathrm{err}_P(g) \leq \widehat{\mathrm{err}}_S(g) + \sqrt{\frac{2\log\widehat{\Pi}_S(\widehat{M}_S(f))}{m}} + 4\sqrt{\frac{2\log(4/\delta)}{m}}. \tag{11}$$

Unlike in the proof of Theorem 1, here the empirical minimal function space $\widehat{M}_S(f)$ is a function of $S$ as well. However, given the fact that $\widehat{M}_S(f)$ also depends on the function $f$, indicates that $\widehat{M}_S(f)$ can still be triggered from any choice of $S$, subject to the choice of the classifying function in $f$. Furthermore, as $\widehat{M}_S(f)$ can fit any set of $m$ sampled points from $P(S)$ (constraint in Section 3 of the main paper), even if $f$ were to be chosen such that it has zero error on the training data samples, it would not change the expected generalization gap for $\widehat{M}_S(f)$. This shows that (11) would hold for $g = f$, leading to the following result.

For $0 < \delta < 1$, with probability $p \geq 1 - \delta$ over the draw of $S$, we have for any function $f$,

$$\mathrm{err}_P(f) \leq \widehat{\mathrm{err}}_S(f) + \sqrt{\frac{2\log\widehat{\Pi}_S(\widehat{M}_S(f))}{m}} + 4\sqrt{\frac{2\log(4/\delta)}{m}}. \tag{12}$$

The main result now follows as $\widehat{KG}_S(f) = \frac{\log\widehat{\Pi}_S(\widehat{M}_S(f))}{m}$. $\quad\square$

## 4.2 Proof of Theorem 2 and Corollary 2.1

First, we provide some results that will help in proving Theorem 2.

**Lemma 1.** *Let $a, b, a', b' \in \mathbb{R}$ s.t. $a > b$. Then $a' < b' \Rightarrow (a' - a)^2 + (b' - b)^2 > \frac{(a-b)^2}{2}$.*

*Proof.* Let $a' = a + \alpha$ and $b' = b + \beta$, where $\alpha, \beta \in \mathbb{R}$ Then $a' < b' \Rightarrow \beta - \alpha > a - b$. We have $(a' - a)^2 + (b' - b)^2 = \alpha^2 + \beta^2 = \frac{(\beta - \alpha)^2 + (\alpha + \beta)^2}{2} > \frac{(\beta - \alpha)^2}{2} > \frac{(a-b)^2}{2}$. $\qquad\square$

We use this result to prove the following Proposition, which is then used in the proof of Theorem 2.

**Proposition 1.** *$f$ and $g$ are two binary classifiers. Let $\epsilon_{max}$ satisfy the following: $\frac{\epsilon_{max}^2}{2} = \max_{X \in \mathbb{R}^d}\{(g_0(X) - f_0(X))^2 + (g_1(X) - f_1(X))^2\}$. Then: $|f_0(X) - f_1(X)| > \epsilon_{max} \Rightarrow argmax_{k \in \{0,1\}}\{f_k(X)\} = argmax_{k \in \{0,1\}}\{g_k(X)\}$. In other words, the winner class of $f$ and $g$ on $X$ are the same.*

*Proof.* Let $a = f_0(X), b = f_1(X), a' = g_0(X), b' = g_1(X)$. Suppose, without loss of generality, that $f_0(X) > f_1(X)$. We are also given that $|f_0(X) - f_1(X)| > \epsilon_{max}$. Then if $g_0(X) < g_1(X)$, Lemma 1 then yields : $D(X) = (g_0(X) - f_0(X))^2 + (g_1(X) - f_1(X))^2 > \frac{(f_0(X) - f_1(X))^2}{2} > \frac{\epsilon_{max}^2}{2}$. From the definition of $\epsilon_{max}$, we have $D(X) > \max_{X \in \mathbb{R}^d}\{D(X)\}$, which is a contradiction. Thus $g_0(X) \geq g_1(X)$. In other words, the winner class of $f$ and $g$ on $X$ is the same. $\qquad\square$

**Theorem 2.** *Given the function $f \in \mathcal{F} : \mathbb{R}^d \to \mathbb{R}^2$ which outputs class logits for binary classification. We construct a function space $\mathcal{F}_{small}^1$ such that $\Pi_m(\mathcal{F}_{small}^1) < \Pi_m(\mathcal{F})$ and $\forall g \in \mathcal{F}_{small}^1$, there exists a description $D_g$ such that $\widehat{\Pi}_S\left(\mathcal{F}(D_g)\right) \leq \widehat{\Pi}_S(\mathcal{F}_{small}^1)$. We approximate $f$ via another function $f_{small}^1 \in \mathcal{F}_{small}^1 : \mathbb{R}^d \to \mathbb{R}^2$ and let $\epsilon_{max}$ be such that*

$$\epsilon_{max}^2 / 2 = \max_{X \in \mathbb{R}^d}\|f_{small}^1(X) - f(X)\|^2. \tag{13}$$

*Denote the output probabilities generated from the corresponding logit outputs of $f(X)$ using the softmax operator (temperature $T = 1$), as $P_0(f(X))$ (label 1 output) and $P_1(f(X))$ (label 2 output). Let $0 \leq \delta \leq 1$ be such that*

$$Pr\left(\left|log\left(\frac{P_0(f(X))}{P_1(f(X))}\right)\right| \leq \epsilon_{max}\right) \leq \delta, \tag{14}$$

*when $X$ is drawn from $S$. Then we have,*

$$\widehat{KG}_S(f) \leq \delta \log 2 + \frac{\log \widehat{\Pi}_S\left(\mathcal{F}_{small}^1\right)}{m}, \tag{15}$$

*where $m$ is the number of samples in $S$.*

*Proof.* First, we note that the result in Proposition 1 implies that $Pr\left(\left|log\left(\frac{P_0(f(X))}{P_1(f(X))}\right)\right| \leq \epsilon_{max}\right)$ actually represents the probability that the output category of the function $f$ and the function $f_{small}^1$ can *possibly* be different. That is, if $\left|log\left(\frac{P_0(f(X))}{P_1(f(X))}\right)\right| \leq \epsilon_{max}$, only then the functions $f$ and $f_{small}^1$ can actually have different labels on $X$. Given this, it follows that if the number of datapoints in $S$ where $f$ and $f_{small}^1$ can differ is less than $m\delta$, then one could jointly describe the function $f$ via the function $f_{small}^1$ and the label variables for all the points in $\mathcal{R}^d$ where the labels can differ, out of which at most $m\delta$ points are in $S$. Note that in order to define $\widehat{KG}_S(f)$, this joint description should fit any $m$ points sampled from $P^m$. This leads to an upper bound for the empirical Kolmogorov growth, given that the number of datapoints in $S$ where $f$ and $f_{small}^1$ can differ is less than $m\delta$, as follows.

$$\widehat{KG}_S(f) = \min_i \frac{\log \widehat{\Pi}_S\left(\mathcal{F}(D_f^i)\right)}{m} \tag{16}$$

$$\leq \frac{\log \widehat{\Pi}_S(\mathcal{F}_{small}^1)2^{m\delta}}{m} \tag{17}$$

$$= \delta \log 2 + \frac{\log \widehat{\Pi}_S\left(\mathcal{F}_{small}^1\right)}{m}. \tag{18}$$

The above holds, only when the number of datapoints in $S$ where $f$ and $f^1_{small}$ can differ is less than $m\delta$, which is the case here, as the expression $Pr\left(\left|log\left(\frac{P_0(f(X))}{P_1(f(X))}\right)\right| \leq \epsilon_{max}\right) \leq \delta$ holds true for $X$ drawn from $S$. $\qquad\qquad\qquad\qquad\qquad\qquad\qquad\qquad\qquad\qquad\qquad\qquad\square$

**Corollary 2.1.** *We are given the function $f : \mathbb{R}^d \to \mathbb{R}^2$ which outputs class logits for the binary classification problem. We construct a set of $K$ function spaces $\mathcal{F}^1_{small}, \mathcal{F}^2_{small}, \mathcal{F}^3_{small}, ..., \mathcal{F}^K_{small}$ using classifiers of decreasing complexity. Thus, we have $\Pi_m(\mathcal{F}^1_{small}) > \Pi_m(\mathcal{F}^2_{small}) > \cdots > \Pi_m(\mathcal{F}^K_{small})$. We also have that $\forall g \in \mathcal{F}^j_{small}$, there exists a description $D_g$ such that $\widehat{\Pi}_S\left(\mathcal{F}(D_g)\right) \leq \widehat{\Pi}_S(\mathcal{F}^j_{small})$, for $j = 1, 2, .., K$. To estimate $\widehat{KG}_S(f)$ using these function spaces, first $f$ is approximated using a function $f^1_{small} : \mathbb{R}^d \to \mathbb{R}^2$, from $\mathcal{F}^1_{small}$ (i.e., $f^1_{small} \in \mathcal{F}^1_{small}$). Then, we continue in a recursive manner, by approximating $f^1_{small}$ using another function $f^2_{small} \in \mathcal{F}^2_{small}$, and so on. Given this, let*

$$\epsilon^1_{max} = \max_{X \in \mathbb{R}^d} \|f(X) - f^1_{small}(X)\|^2 \tag{19}$$

$$\epsilon^2_{max} = \max_{X \in \mathbb{R}^d} \|f^1_{small}(X) - f^2_{small}(X)\|^2 \tag{20}$$

$$\dots$$

$$\epsilon^K_{max} = \max_{X \in \mathbb{R}^d} \|f^{K-1}_{small}(X) - f^K_{small}(X)\|^2. \tag{21}$$

*Denote the output probabilities generated from the corresponding logit outputs of $f(X)$ using the softmax operator (temperature $T = 1$), as $P_0(f(X))$ (label 1 output) and $P_1(f(X))$ (label 2 output). Now, let real constants $0 \leq \delta_1, \delta_2, ..., \delta_K \leq 1$ be such that*

$$Pr\left(\left|log\left(\frac{P_0(f(X))}{P_1(f(X))}\right)\right| \leq \epsilon_{max}\right) \leq \delta_1, \tag{22}$$

$$Pr\left(\left|log\left(\frac{P_0(f^1_{small}(X))}{P_1(f^1_{small}(X))}\right)\right| \leq \epsilon^2_{max}\right) \leq \delta_2, \tag{23}$$

$$\dots$$

$$Pr\left(\left|log\left(\frac{P_0(f^{K-1}_{small}(X))}{P_1(f^{K-1}_{small}(X))}\right)\right| \leq \epsilon^K_{max}\right) \leq \delta_K, \tag{24}$$

*given that $X$ is drawn from $S$. Then we have,*

$$\widehat{KG}_S(f) \leq \sum_{i=1}^{K} \delta_i \log 2 + \frac{\log \widehat{\Pi}_S\left(\mathcal{F}^K_{small}\right)}{m}, \tag{25}$$

*where $m$ is the number of samples in $S$.*

*Proof.* First, proceeding similarly to the proof of Theorem 2, we note that a tighter bound on $\widehat{KG}_S(f)$ can be achieved by realizing that $f^1_{small}$ doesn't have to belong to $\mathcal{F}^1_{small}$, but can belong to its empirical minimal function space $\widehat{M}_S(f^1_{small})$ itself, which leads to

$$\widehat{KG}_S(f) = \min_i \frac{\log \widehat{\Pi}_S\left(\mathcal{F}(D^i_f)\right)}{m} \tag{26}$$

$$\leq \frac{\log \widehat{\Pi}_S\left(\widehat{M}_S(f^1_{small})\right) 2^{m\delta}}{m} \tag{27}$$

$$= \delta_1 \log 2 + \frac{\log \widehat{\Pi}_S\left(\widehat{M}_S(f^1_{small})\right)}{m} \tag{28}$$

$$= \delta_1 \log 2 + \widehat{KG}_S(f^1_{small}) \tag{29}$$

The proof of this corollary then trivially follows by noting that one could recursively approximate $f^1_{small}$ by $f^2_{small}$ to yield

$$\widehat{KG}_S(f^1_{small}) \leq \delta_2 \log 2 + \widehat{KG}_S(f^2_{small}), \tag{30}$$

and similarly for the rest. For the final approximation, we use the function space $\mathcal{F}^K_{small}$ to yield the final upper bound as

$$\widehat{KG}_S(f^{K-1}_{small}) \leq \delta_K \log 2 + \frac{\log \widehat{\Pi}_S \left( \mathcal{F}^K_{small} \right)}{m}. \tag{31}$$

The result in the corollary then follows by combining all bounds. $\qquad\square$