# OpenReview forum: "Network-to-Network Regularization: Enforcing Occam's Razor to Improve Generalization"
_NeurIPS.cc/2021/Conference — NeurIPS 2021 Poster_

### Official Review · Reviewer_muBt · 2021-07-11

**Rating:** 7
**Confidence:** 3

**Summary:**

The paper proposes a novel and theoretically inspired regularization techniques for training deep neural networks. The technique involves regulating the network by regressing the output of the model to a simpler model but the simpler model also regresses towards the output of the base model. The experimental results show that the proposed techniques are effective at reducing the generalization gap and improving performance in presence of label noise.

**Main Review:**

The paper first defines Kolmogorov Growth (KG) which is a variation of growth function that in addition considers the description length of the model that can still fit the data. Since KG is a growth function, the generalization gap can easily be bounded via Massart's finite lemma (Thm 1). The paper then proceeds to show that if a smaller model can fit the predictions of the base model, then KG of the larger model can be bounded by the growth function of the smaller model (Thm 2). The proof of theorem 1 is relatively straightforward but I have some questions regarding theorem 2 which I will elaborate below.

Regarding the proposed method, the method seems fairly novel and is well motivated by the proposed theorem. The results show that it offers small improvement on standard classification benchmark and significant improvement in the presence of label noise which is pretty cool.

Now I will discuss the concerns I have regarding the algorithm and experimental results:

- The paper does not describe the motivation behind using multiple levels of distillation of increasingly small capacities. While the theorem shows that you can bound the KG by chaining inequality, it does not explain why this is necessary. Empirically it seems there are some benefits but it is unclear why we need multiple levels. Afterall, bounding with one level (the smallest) will yield tighter bounds. While this might be hard to rigorously justify, I want to see more discussion about intuition and if possible theoretical results.
-nEmpirically, how does the method compare to reverse distillation?
- Have you verified the conditions of theorem 2? To me it seems this condition is impossible to verify thus you don’t really know whether you are bounding the KG. In other words, did you compute $\delta$ in addition to the growth function? For example, I can pick a really small network so the $\delta$ is large -- Now the KG is dominated by $\delta$ instead of the growth function.
- Following the previous point, does the smallest student model fit the label perfectly? In my experience, models with 11k/16k parameters can have a hard time fitting cifar10 / cifar100 perfectly. If they don’t then the theory of KG breaks down.
- How did you compute the growth function for neural networks for the RHS of eq 9, thm 2? Even small networks should have very high growth functions. As far as I can tell it’s also intractable to compute the exact growth function of an arbitrary neural network. If you relied on approximation then what method did you use?
- The method has some hyperparameters. Could you discuss strategies for choosing the hyperparameters?

**Questions about proof**:

I have some questions about the details of thm 2’s proof. It is not immediately obvious to me that the conditions described by the first two sentences of the proof are true. Could you add more details to that part of the proof seeing that it is possibly the most important theoretical result of the paper.

I will use $g$ to denote $f_{small}^1$.

First we can rewrite $$\log\left( \frac{P_0(f(X)}{P_1(f(X))} \right) = f_0(X) - f_1(X)$$ and $$ \epsilon_{max} = \max_{X \in \mathbb{R}^d} || g(X) - f(X) ||^2 = \max_{X \in \mathbb{R}^d} (g_0(X) - f_0(X))^2 + (g_1(X) - f_1(X))^2$$

The two sentences are essentially saying that $$|f_0(X) - f_1(X)| \leq \epsilon_{max} \Longrightarrow argmax_{k} f_k(X) \ne  argmax_{k}  g_k(X)$$. Could you elaborate why this is true?


**Other comments**:

- Equation 9 missing hat?
- The related work section of the work seems kind of sparse as generalization for deep learning has attracted a lot of attention recently. I listed a few works at the end of this review that I think you can consider to discuss.

Overall, I think this is a good paper with a nice empirical method motivated by principled theory. If the authors can address my concerns, I would be happy to raise my score to 7.

**Reference**

[1] Uniform convergence may be unable to explain generalization in deep learning. Nagarajan et al.

[2] Towards Learning Convolutions from Scratch. Neyshabur et al.

[3] Fantastic Generalization Measures and Where to Find Them. Jiang et al.

[4] In Search of Robust Measures of Generalization. Dziguite et al.

[5] Transferring Inductive Biases through Knowledge Distillation. Abnar et al.

I am not entirely up-to-date with literature on distillation so I could have missed some there.


**Time Spent Reviewing:**

6

---

> ### Author Response · Authors · 2021-08-10
> **Response**
>
> We thank the reviewer for their detailed comments and insights. The following are our clarifications to the comments and questions. All relevant changes, results and discussions will be added to the revised version of the paper.
>
> 1. From an empirical viewpoint, as seen in our reported results, our finding has been that multi-level N2N yields better generalization error than single level N2N. The intuitive explanation for this phenomenon is that the approximation loss terms (mean-squared error based) between the networks in the multi-level setup are more tightly bounded (individual losses will be smaller) which allows for more efficient training. But a rigorous explanation of this fact is indeed currently lacking, and is a subject for further investigation and research.
> 2. We’ve since run experiments with reverse distillation (RD) in a like-for-like setting (same network architecture), for both the no label noise and the label noise cases. Overall we find that for the no label noise case, 2-level N2N is better or roughly similar in performance to RD across the three datasets, with a decrease in error rate within the range of 5-18\%. For the label noise case however, we see greater relative improvement in test error of N2N (up to 62\% lower test error rate for symmetric noise and 90\% lower error rate for asymmetric pair-wise noise). This is because in the case of RD, the smaller network is already pre-trained to fit the labels on the given dataset, and thus the larger network doesn’t learn too differently from the normal case. Whereas in N2N, we only constrain the base network to stay “near” some smaller network configuration, so there is no direct supervision performed on the smaller network, which is an important differentiator.
> 3. Yes, we indeed computed the $\delta$ term for all the experiments which reported the bound on KG. In fact, all changes in the KG bound in the plots that we see in Figure 2 (a) and (b) are changes in $\delta$, rather than the growth function, as the growth function term is fixed because of the smaller network. Our main objective with Theorems 1 and 2 was to develop an error bound that could change based on the final choice of classification function, which is primarily reflected in the $\delta$ term of Theorem 2.  The growth function term is fixed and does not respond to the base network function choice, and is reminiscent of the traditional error bounds which contain complexity terms based on entire function spaces.
> 4. Yes, our experience is that in CIFAR-10/100, the smaller networks do not reach zero training error at all. The result of Theorem 1 indeed rests on this assumption that the smaller network function space must be able to perfectly fit the data. For Theorem 2 however, there is no such assumption. To elaborate, in Theorem 2, the $\delta$ term is primarily reflective of the extent to which the smaller network can approximate the base network. As such, if a smaller network’s function space wouldn’t be able to exactly fit the labels, it would have non-zero $\delta$, and the empirical KG of the base network function would depend on the value of approximation error in $\delta$, in addition to growth function of the smaller network.
> 5. We clarify that we were able to approximate the trained base network function with 2-layer CNNs ($\mathcal{F}^2_{small}$) having as low as only 504 trainable parameters (with negligible deltas in Theorem 2 equation (9)), for the KG bound computation on MNIST. Subsequently, the well-known VC dimension based approximation ([1], Jiang et. al, 2020) was used to bound the growth function of the $\mathcal{F}^2_{small}$ CNN. Note that once computed, the growth function for all experiments in Figure 2 (a) and (b) is unchanged, as $\mathcal{F}^2_{small}$ is fixed for all experiments reported in Figure 2.
> 6. For both the no label noise and label noise experiments, we tuned the $\lambda$ hyperparameters using a grid search over a fixed validation set, after which, we use the entire training data to train the networks with the chosen optimal $\lambda$ parameters. For selecting the smaller network configurations on MNIST, we used the validation set to decide the best network configuration out of a set of possible architectures we had created beforehand (2/3/4 layer CNNs with variable width). For the experiments on the CIFAR-10 and CIFAR-100 datasets, we simply gradually scaled the smaller networks optimized for MNIST by their width (after adding an fc layer), and used them as the network configurations for all experiments. Regularization parameter details and network architecture details are provided in the supplementary material.
>
> 7. Proof of Theorem 2 Clarification:
> We apologise for the language in the first two sentences of the proof of Theorem 2, which may have led to a mis-interpretation of what we intended to say. In the reviewer’s notation, we would like to clarify that the first two sentences of the proof are essentially saying the following:
> $|f_0(X)-f_1(X)|>\epsilon_{max} \Rightarrow argmax_k{f_k(X)}=argmax_k{g_k(X)}$.
> In other words, if the absolute difference in logits of the function $f$ on $X$ is greater than a particular threshold (which depends on the max squared-difference between $f$ and $g$ over all $X$), then the output “winning” category of $f(X)$ and $g(X)$ will be the same, since $||f(X)-g(X)||^2$ would have to be greater than its maximum value to overturn the winning category in $g(X)$.
> We also note that we changed equation (13) in the supplementary material by re-defining $\epsilon_{max}$ to satisfy the following:
> $ \epsilon_{max}^2/2 = \max_{X \in \mathbb{R}^d} \lVert f^1_{small}(X) - f(X) \rVert^2$
>
> Other comments: We have fixed equation (9) and we will update the related work section with the suggested references.
>
>
> References:
> [1] Yiding Jiang, Behnam Neyshabur, Hossein Mobahi, Dilip Krishnan, Samy Bengio, “Fantastic Generalization Measures and Where to Find Them”.

---

> > ### Comment · Reviewer_muBt · 2021-08-29
> > **Increasing score**
> >
> > I thank the authors for the detailed response.
> > I believe most of my concerns have been addressed and thus I am increasing my score to 7.
> >
> > There is one problem left regarding to theorem 2. I believe the theorem is true but it is still not clear how the exact derivation is done even after your clarification. The fact that you have to change the definition of $\epsilon_{max}$ suggests that the argument was not that straightforward to begin with. Please make it rigorous.

---

> > > ### Author Response · Authors · 2021-09-01
> > > **Response**
> > >
> > > We thank the reviewer for the positive feedback. The following is a detailed explanation of the first part of the proof of Theorem 2, namely the following claim.
> > >
> > > *Claim:* $f$ and $g$ are two binary classifiers. Let $\epsilon_{max}$ satisfy the following: $\frac{\epsilon_{max}^2}{2}=\max_{X\in \mathbb{R}^d}\\{(g_0(X)-f_0(X))^2+(g_1(X)-f_1(X))^2\\}$.
> > >
> > > Then:
> > > $|f_0(X)-f_1(X)|>\epsilon_{max} \Rightarrow argmax_{k\in \\{ 0,1 \\} }{f_k(X)}= argmax_{k\in\\{ 0,1 \\} }{g_k(X)}$.
> > >
> > > In other words, the winning category of $f$ and $g$ on $X$ are the same.
> > >
> > > Before we prove the claim, consider the following proposition.
> > >
> > > *Proposition:* Let $a,b,a',b'\in\mathbb{R}$ s.t. $a>b$. Then $a'<b'\Rightarrow (a'-a)^2+(b'-b)^2>\frac{(a-b)^2}{2}$.
> > >
> > > *Proof of Proposition:* Let $a'=a+\alpha$ and $b'=b+\beta$, where $\alpha,\beta\in\mathbb{R}$. Then $a'<b'\Rightarrow \beta-\alpha>a-b$. We have $(a'-a)^2+(b'-b)^2=\alpha^2+\beta^2=\frac{(\beta-\alpha)^2+(\alpha+\beta)^2}{2}\geq\frac{(\beta-\alpha)^2}{2}>\frac{(a-b)^2}{2}$. QED.
> > >
> > > *Proof of Claim:*
> > > Let $a=f_0(X),b=f_1(X),a'=g_0(X),b'=g_1(X)$.
> > > Suppose, without loss of generality, that $f_0(X)>f_1(X)$. We are also given that $|f_0(X)-f_1(X)|>\epsilon_{max}$. Then if $g_0(X)<g_1(X)$, the proposition gives us: $D(X)=(g_0(X)-f_0(X))^2+(g_1(X)-f_1(X))^2>\frac{(f_0(X)-f_1(X))^2}{2}>\frac{\epsilon_{max}^2}{2}$. From the definition of $\epsilon_{max}$, we have $D(X)>\max_{X\in \mathbb{R}^d}\\{D(X)\\}$, which is a contradiction. Thus $g_0(X)\geq g_1(X)$. In other words, the winning category of $f$ and $g$ on $X$ is the same. QED.

---

> > > > ### Comment · Reviewer_muBt · 2021-09-01
> > > > **Response**
> > > >
> > > > This looks good. Please make sure this and other changes are incorporated in the final version if the paper is accepted.

---

### Official Review · Reviewer_NwZ7 · 2021-07-16

**Rating:** 7
**Confidence:** 4

**Summary:**

This paper proposes a measure of function complexity that is based on a function itself rather than the function class it belongs to. It uses this complexity measure to establish generalization upper bounds for classifiers. In addition, an empirical method for approximating this measure, and optimizing it, is proposed and evaluated on three small benchmark datasets, exhibiting slightly improved test accuracy over existing baselines. The method is also shown to improve resilience to label noise.

**Limitations And Societal Impact:**

The authors don't seem to have provided a section on negative societal impacts. In the checklist they actually only mention a possible positive societal impact. On the other hand, I fail to see any potential significant negative societal impacts specific to this work. Therefore, I would not consider this to be a mark against the paper.

As for limitations, the only thing that was mentioned was the fact that the hyperparameters must be manually tuned. The limitations could certainly be expanded upon.

**Main Review:**

The problem addressed is significant, as indeed most function class-based generalization bounds are not particularly useful for deep networks. The work seems to be original, and is fairly clear. The theoretical analysis seems reasonable, and the experiments show slight improvements over baselines; however, the amount of improvement is not substantial.

Questions:

Are the KG-based generalization bounds (empirically approximated using the provided method) actually useful? It would be good to compute the right hand side of Corollary 1.1 (with KG_S replaced by the bound in (9)) on the datasets and compare this to the test error.

Do the same trends in Table 2 hold up when the label noise is not symmetric?

Areas for improvement:

It would be interesting to compare the empirical KG of different model architectures. For example, do modern CV architectures (e.g., ResNets, DenseNets) have a lower empirical KG than older-style architectures (from simple fully-connected networks to VGG)? In my view, this would substantially strengthen the paper as an additional way to support the usefulness of the metric.

The authors should discuss and cite "Fantastic Generalization Measures and Where to Find Them" (Jiang et al., 2019), which has a more exhaustive list of previously proposed complexity measures. In addition, this paper has strong empirical evaluations of these measures; adding in some of these evaluations as appropriate (and showing that empirical KG outperforms other existing complexity measures on these metrics) would also strengthen the paper.

The authors should include training time results (e.g., wall-clock time). The proposed N2N method is more expensive than standard training as it involves training multiple networks. In terms of total parameter count, I am not sure whether the N2N approach yields better accuracy than a slightly larger network with total number of parameters approximately equal to the sum of the number of parameters in the base and smaller networks in N2N (especially given the relatively small size of the improvements).

These evaluations could cause me to raise my score.

Minor comments:

The related work section 5.2 is somewhat lacking. Although the discussion of reverse knowledge distillation is good, more related work can be discussed. (However, a fair amount of related work is discussed in the introduction.)

The tables can be tough to read. Increasing the line spacing slightly would be good.

There are multiple grammar errors (such as extraneous commas); I recommend that the authors carefully copyedit before the final version.

**Time Spent Reviewing:**

6

---

> ### Author Response · Authors · 2021-08-10
> **Response**
>
> We thank the reviewer for their detailed comments and useful suggestions. The following are our clarifications to the comments and questions. All relevant discussions and results will be added to the revised version of the paper.
>
> Questions:
> 1. N2N regularization reduces the KG bound, but we still find that the resulting KG bound itself can be quite loose compared to the true generalization gap (primarily due to the growth function term of the smaller network in equation (9)). However, in experiments in Figure 2, and in other experiments since, we are consistently seeing that lower/higher values of the bound have a strong causal relationship with the resulting generalization gap (average correlation of ~0.94 within each dataset). We find that any form of regularization (dropout/l2-norm included) consistently yields networks of lower KG (by 5-10\%), and also agrees with the improved generalization gap in these datasets. A possible direction of future work is to impose additional structural constraints (e.g., equivariant constraints such as rotation and scale) to yield very low complexity student networks to yield tighter bounds.
> 2. Yes, the same trends do hold for asymmetric label noise. We just tested our methods in the case when the noise was asymmetric (pairwise noise of 45\%, similar to [1] Han et. al. 2018). On MNIST, by using networks of approximately half-parameter size as the smaller networks (12K params for level 1, 3.3K params for level 2) for the proposed N2N algorithm, we find the same trend in results, i.e. N2N performs significantly better than the other reported methods. We’re also currently running experiments on the CIFAR-10 and CIFAR-100 dataset, and all results will be added to the paper or the supplementary material, subject to available space.
>
> Areas for Improvement:
> 1. We tested this out in the case of CIFAR-10, and the results were interesting. We compared two Resnet configurations (Resnet-18 and Resnet-44) against a VGG-9 and VGG-16 architecture. We find that the KG of the trained Resnets are consistently lower than the trained VGG architectures by a value within the range of 0.01-0.03 (10-20\% reduction in $\delta$ in Theorem 2). We note that this is in spite of the fact that the Resnet architectures have a greater parametric count (11M for Resnet-18 versus 0.5M for VGG-9) and larger depth.
> 2. We will cite the paper and discuss the relevance of the metrics therein to the proposed KG metric. However, we feel that given the additional set of analysis we’re already adding to the paper, such a large-scale evaluation of performance between various complexity metrics is out of scope for this work. We will be treating this as a potential extension of this work.
> 3. (i) Training Time: We’ve now added the training time results to the paper. On average, we’re seeing slower training by a factor of 1.4-1.5 in all three datasets.
> (ii) Model Complexity: We concur that the total number of parameters involved in the entire training process is larger, but we note that the relative increase is quite small compared to the number of parameters in the base model (for e.g. 0.6M versus 0.58M for MNIST). We tested out networks with these slightly larger parametric counts (expanding width) on MNIST and CIFAR-10, and found that the improvement in the base network’s performance was much lower compared to the improvement from the N2N algorithm. We note that even though the total number of parameters during training is larger than the base network, at test time, only the base network is used for generating class decisions. This is similar to what happens in knowledge distillation.
> 4. We will address all minor comments accordingly. Limitations will be expanded upon with relevant discussions.
>
> References
>
> [1] Han, Bo and Yao, Quanming and Yu, Xingrui and Niu, Gang and Xu, Miao and Hu, Weihua and Tsang, Ivor and Sugiyama, Masashi, ”Co-teaching: Robust training of deep neural networks with extremely noisy labels” Neurips 2018.

---

> > ### Comment · Reviewer_NwZ7 · 2021-09-01
> > **Thank you for the response**
> >
> > Some follow-up questions/comments:
> >
> > 1. For the ResNet/VGG results in the above reply, was the KG estimated using a single student or the recursive approach?
> > 2. In the final paper, it would be good to include KG estimates corresponding to the settings in Tables 1-2 to confirm that higher accuracies are correlated with lower KGs.
> > 3. What was the reason that a ResNet-44 was used for CIFAR-10 rather than a ResNet-50? ResNet-50 with L2 regularization should be able to achieve an accuracy that actually exceeds the 93.35 figure reported in the paper for ResNet-44 with drop+l2+N2N. The CIFAR-10 result would thus be more convincing if the same improvements still hold up on a ResNet-50 (since it is harder to improve the performance of models that are already better). I do not expect the authors to include this result during the response period given the limited time remaining, but it would be useful for the final version.
> > 4. Is the assumption made that the true label is a deterministic function of the example? I.e., is it assumed that some determinstic $g_{true}$ does exist (in other words, the example contains enough information for $g_{true}$ to always determine the label with certainty).
> > 5. The fact that a VC dimension based approximation is used to bound the growth function of the smaller network (mentioned in reply to reviewer muBt) is certainly a key detail and should be mentioned in the paper.
> >
> > Minor:
> > 1. In (4), $err_P(f, g_{true})$ is defined as $\lim\limits_{N\rightarrow\infty, z_i \sim P} \sum\limits_{i=1}^N \dfrac{(1-f(z_i)g_{true}(z_i))}{2N}$. Why not just write $E_{z_i \sim P} \left[ \dfrac{1-f(z_i)g_{true}(z_i)}{2} \right]$ instead?
> > 2. It would be better to use the standard notation (x_i, y_i) rather than (z_i, y_i) to denote an (example, label) pair (unless there is a specific reason for the latter that I missed).

---

> > > ### Author Response · Authors · 2021-09-01
> > > **Response**
> > >
> > > We thank the reviewer for the follow-up queries and suggestions. Here are our responses to the same. All changes will appear in the final version of the paper.
> > >
> > > 1. We used the single-student approach (2-layer CNN) to estimate the KG of the VGG/ResNet networks.
> > > 2. Yes, we will include the estimated KG bounds in the tables to highlight the correlation between test error and KG in the final version of the paper.
> > > 3. We found that Resnet-44 was easier to optimize for the CIFAR-10 dataset, as it has significantly fewer parameters compared to ResNet-50 (0.66M v/s 23M). Our objective was mainly to demonstrate that N2N could improve performance of competitive ResNets and hence ResNet-44 seemed like a good choice. As suggested, we will explore the performance of ResNet-50 on CIFAR-10.
> > > 4. We do not make the assumption that $g_{true}$ is a deterministic function of the input data. No assumption about $g_{true}$ is made.
> > > 5. Yes, we will mention this in the final version of the paper.
> > > 6. For the minor comments, we will make the relevant changes accordingly.

---

### Official Review · Reviewer_aPxa · 2021-07-16

**Rating:** 7
**Confidence:** 3

**Summary:**

This paper proposes a Kolmogorov Growth (KG), a novel measure of the complexity of a neural network. They derive new generalization bounds using KG and take inspiration from this bound to propose a new way of regularizing neural networks (network-to-network regularization, N2N). Finally, the paper verifies their regularization scheme on three standard image benchmarks.

**Limitations And Societal Impact:**

Please refer to the main review.

**Main Review:**

As reflected in its name, Kolmogorov Growth takes inspiration from Kolmogorov complexity and the growth function. Conceptually, KG measures the size of the simplest class of functions that a given function f belongs to. The main innovation seems to be the min_i operator, which restricts our attention to the simplest function class. KG by itself is obviously not computable, but the paper proposes an interesting way to approximate it.

The generalization bound in section 3.1 is not particularly surprising. In a sense, the proof for this bound is embedded in the definition of KG itself. Still, I think this is an interesting approach to bounding the generalization error.

Network-to-network regularization is motivated by the results in section 3.1, which show that functions that can be approximated well by small networks will have low KG. N2N trains a small student network and a large teacher network to output similar predictions to each other. This makes the student learn from the teacher, while the teacher is regularized by the simpler function of the student. The paper also considers a multi-level version where a sequence of progressively smaller networks are trained with the same loss.

N2N hinges on the implicit assumption that smaller networks have smaller hypothesis spaces and thus learn simpler functions. While this is likely true in a strict sense, many results with neural networks show that small networks have surprisingly strong expressive power. For example, works on knowledge distillation show that minimal networks can achieve accuracy close to that of a large teacher, and [1] shows that a rather small network can fit random labels. I think this implicit assumption and its limitations should be discussed a bit more in the paper.

Experiments show that N2N acts as an effective regularizer and that N2N regularization indeed reduces the KG bound. It would have been interesting to additionally include the generalization gap in Figure 2 so that the KG bound can be compared with it.

There may be a more direct correspondence between Kolmogorov complexity and KG. In KG’s definition (section 3), you essentially construct a short two-part program for describing the function f where you first describe the function class F and then specify f within F. I wouldn’t be surprised if the two concepts can be shown to have a stronger relation than KC simply acting as “indirect motivation” for KG (line 140). I think this is an exciting direction for future work.

[1] Understanding deep learning requires rethinking generalization, Chiyuan Zhang, Samy Bengio, Moritz Hardt, Benjamin Recht, Oriol Vinyals


**Time Spent Reviewing:**

6

---

> ### Author Response · Authors · 2021-08-10
> **Response**
>
> We thank the reviewer for their useful comments and insights. The following are our clarifications to the comments. Relevant discussions and results will be added to the revised version of the paper.
>
> 1. This paper makes the implicit assumption that smaller networks have smaller hypothesis spaces. This indeed does not mean that they can only model simple functions and we agree that smaller networks can have high expressive power. However, that power is only accessed when the network is forced to train on a dataset with random labels (i.e., high complexity ground truth function), as we see in Figure 2 of our paper. It was also observed in ([1] Valle-Perez et. al. 2018), that a random choice of weights on the smaller network will likely yield a function of low Kolmogorov complexity. This essentially implies that most functions resulting from random assignments of weight values on the smaller network will have a lower KG as well, a fact that we can demonstrate empirically.
> 2. N2N regularization targets reducing the KG bound, but we still find that the resulting KG bound itself can be quite loose compared to the true generalization gap (primarily due to the growth function term of the smaller network in equation (9)). However, in experiments in Figure 2, and in other experiments since, we are consistently seeing that lower/higher values of the bound have a strong causal relationship with the resulting generalization gap (average correlation of 0.94 within each dataset). We find that any form of regularization (dropout/l2-norm included) consistently yields networks of lower KG (by 5-10\%), and also agrees with the improved generalization gap in these datasets. A possible direction of future work is to impose additional structural constraints (e.g., equivariant constraints such as rotation and scale) to yield very low complexity student networks to yield tighter bounds.
> 3. We agree that the KC and KG may be more intimately related, beyond the indirect motivation we mentioned. It is intuitively clear that functions of lower KG will have lower KC and vice-versa. We are planning to explore this direction. Thank you for the inspiration!
>
> References
>
> [1] Guillermo Valle-Pérez, Chico Q. Camargo, Ard A. Louis, “Deep learning generalizes because the parameter-function map is biased towards simple functions”, ICLR 2019.

---

### Official Review · Reviewer_LCjW · 2021-07-19

**Rating:** 6
**Confidence:** 4

**Summary:**

This paper addresses the fundamental problem of the generalization ability of deep neural networks and aims to shed light on the theoretical aspects of their generalization ability by introducing a approach based on a new complexity measure called "Kolmogorov Growth" (KG). A practical neural network training approach called Network-to-Network (N2N) Regularization is also introduced with the aim of enforcing the low KG condition so that the generalization gap is reduced. Theoretical results are accompanied by relevant experimental evaluations, including a noisy-label case investigation, which helps in validating the proposed KG-based N2N training of neural networks. Overall, the material introduced is seemingly novel, and although tries to address a well-established problem, the results presented are good guidance for research extensions in this direction.

**Ethical Concerns:**

There are no immediate ethical concerns or issues with this paper.

**Ethics Review Area:**

["I don’t know"]

**Limitations And Societal Impact:**

The limitations of this work are briefly discussed in the Conclusion section (Sec. 7). However, a more complete discussion on the limitations and how to overcome them in future research is warranted.

There is no discussion in the paper on the potential negative societal impact of this work.

**Main Review:**

Merits:
1. Overall, the paper is well-written and theoretically sound. The idea of extending the concept of Growth Function to measure the descriptional complexity of a function space and connecting it with the well-known Rademacher complexity-based Generalization gap theorem from Learning Theory is new, and of interest to the research community.
2. Although the theoretical results presented themselves are not entirely original and largely inspired from existing gen. gap results, coming up with the KG measure and developing a measurable approximation based on the recursive student classifier approach is still novel.
3. The experimental evaluation is quite thorough, and attempts to validate most aspects of the theoretical results. The impact of the properties of the smaller networks (e.g. rotational invariance for CNNs) on the larger network is also briefly analyzed, which would be interesting for further research in this direction.

However, I do have  the following questions and comments.

Questions/ Comments:
1. In Algorithm 1 table, the first line reads ``Training data $\{X_i, y_i \}_{i=1}^k$''. What is $k$ here? Do you mean $m$ instead of $k$, where $m$ is the number of training samples?
2. In Algo 1 table, what is $e_{base}$ and $e_{small}$? How is $e_{base}$ different from $J$? Later in Sec 6.1, it is mentioned that $e_{base}=3$ and $e_{small}=1$, but in Algo 1, $X_i$ and $y_i$ is used throughout, which does not make sense. The overall notation and usage in Algorithm 1 needs much more clarification.
3. In Sec 6.1, it is said that "larger training data sizes need smaller regularization parameters". Could the authors provide any more insight into this observation, especially in the light of the introduced KG?
4. In Sec 6.1, you say "2-level N2N regularization" and then mention $m=2$, but this makes no sense. Do you mean $K=2$, i.e., 2 smaller networks trained. Similarly with single-level N2N?
5. In Sec 6.2, the observation that a larger $\lambda_0$ gives a higher test accuracy. Could the authors provide any intuitive explanation for this trend, as to why KG decreases with higher $\lambda_0$? In Fig. 2, what is $K$, i.e., how many student networks are used to learn the original teacher network?

**Time Spent Reviewing:**

4

---

> ### Author Response · Authors · 2021-08-10
> **Response**
>
> We thank the reviewer for their useful comments and suggestions. The following are our clarifications to the comments and questions. All changes will appear in the revised version of the paper.
>
> 1. Yes, that is an error on our part, we meant $m$ instead of $k$.
> 2. Let us clarify the algorithm flow and notation. The algorithm trains the base network and the smaller networks in an alternating manner. First the base network is trained for $e_{base}$ epochs. Then the smaller networks are trained sequentially, each for $e_{small}$ epochs. This procedure is repeated $J$ times. Also, the conflict of notation within the algorithm w.r.t the variable "$i$" has been fixed.
> 3. Empirically we observed that for larger training data sizes smaller values of $\lambda$ yielded the best results. Our explanation for this is as follows. As the number of training data samples increases, over-parameterized models will need less regularization, as the training data points are dense enough for the model to efficiently approximate the underlying ground truth label generating function. Our empirical results reinforce that minimizing KG via the N2N approach is a form of regularization.
> 4. That is a typo. It should be: $K=2$ and $K=1$ instead of $m=2$ and $m=1$, respectively.
> 5. (i) KG decreases with higher $\lambda$: As $\lambda_0$ increases, the loss function focuses more on minimizing the N2N approximation error, which leads to network configurations which are more easily approximable using smaller networks. This implies a smaller upper bound on KG due to a smaller value of the $\delta$ term in Theorem 2 (recall that $\delta$ is related to the approximation error).
> (ii) Larger $\lambda_0$ yields better accuracy: As $\lambda_0$ increases, more emphasis is put on lowering KG, which improves generalization and yields better test accuracy. However, this happens up to a threshold and test accuracy decreases as $\lambda_0$ increases beyond the threshold. In the case of training data with noisy labels, we find that the threshold is larger (see Fig 1) because we can put less emphasis on fitting the noisy training labels and more emphasis on minimizing KG.
> (iii) Fig.2 what is $K$?: In Fig. 2, the number of smaller networks used to train the base network is $K=2$. We show results for varying $\lambda_0$ and $\lambda_1=0.1$.
> 6. We will expand on the limitations of our work and add relevant discussion to the paper.

---

> > ### Comment · Reviewer_LCjW · 2021-08-30
> > **Thanks for the response**
> >
> > Thank you authors for addressing my questions and comments. With the understanding that the corrections and changes will be incorporated, I stand by my original score.

---

### Decision · Program_Chairs · 2021-09-27

**Decision:**

Accept (Poster)

**Comment:**

This paper introduced a novel measure of the complexity of a class of functions. Reviewers agreed that the proposal was novel, and theoretically sound. Reviewers also felt that experiments exploring regularizing using (a proxy for) Kolmogorov Growth were convincing. One reviewer increased their score, and another increased their confidence in their (accept) score during discussion, both of which I take as very positive signs.

Reviewers had specific actionable feedback in their reviews, which the authors should incorporate into their manuscript before the camera ready.